# Functional genomic analysis identifies miRNA repertoire regulating *C. elegans* oocyte development

Amanda L. Minogue[1,2], Michael R. Tackett[3], Elnaz Atabakhsh[3], Genesis Tejada[3] & Swathi Arur [1,2]

Oocyte-specific miRNA function remains unclear in mice and worms because loss of Dgcr8 and Dicer from mouse and worm oocytes, respectively, does not yield oogenic defects. These data lead to several models: (a) miRNAs are not generated in oocytes; (b) miRNAs are generated but do not perform an oogenic function; (c) functional oocyte miRNAs are generated in a manner independent of these enzymes. Here, we test these models using a combination of genomic, expression and functional analyses on the *C. elegans* germline. We identify a repertoire of at least twenty-three miRNAs that accumulate in four spatial domains in oocytes. Genetic tests demonstrate that oocyte-expressed miRNAs regulate key oogenic processes within their respective expression domains. Unexpectedly, we find that over half of the oocyte-expressed miRNAs are generated through an unknown Drosha independent mechanism. Thus, a functional miRNA repertoire generated via Drosha dependent and independent pathways regulates *C. elegans* oocyte development.

[1] Program in Developmental Biology, Baylor College of Medicine, Houston, TX 77030, USA. [2] Department of Genetics, U.T. MD Anderson Cancer Center, Houston, TX 77030, USA. [3] Abcam, One Kendall Square, Cambridge, MA 02142, USA. Correspondence and requests for materials should be addressed to S.A. (email: sarur@mdanderson.org)

MicroRNAs (miRNAs) are 21–22 nucleotide noncoding RNAs that regulate gene expression via post-transcriptional repression of target mRNAs[1]. In the canonical pathway, the microprocessor complex, composed of Drosha and Dgcr8 (Pasha), generates pre-miRNAs from primary miRNAs in the nucleus[1]. In the cytoplasm, Dicer generates mature miRNAs from pre-miRNAs. The Argonaute (Ago) family of proteins bind to the mature miRNAs and mediates repression of the specific target RNAs either by RNA degradation or by inhibiting translation[1]. Argonaute proteins achieve specific target inhibition as part of the RNA-induced silencing complex (RISC) where they are guided by the miRNA to complementary target sites in mRNAs. The active Ago proteins then cleave the targets or inhibit translation[2]. Functionally, miRNAs regulate multiple cellular events[3,4], and their mis-regulation is associated with diseases, such as cancer[5–7]. Despite a central role in the development of most tissues[8], a role for miRNAs during oogenesis remains unclear in worms and mice, in part, because loss of Dgcr8 from mouse oocytes[9], or Dicer from *C. elegans* germlines or mouse oocytes[9,10], does not affect oocyte development. Suh et al.[9], analyzed 40 highly expressed miRNAs from Dgcr8 and Dicer deleted GV stage mouse oocytes and found that 39 of the 40 miRNAs were downregulated in both Dgcr8−/− and Dicer−/− oocytes[9]. However, the authors also found that there was no change in gene expression between wild-type and Dgcr8−/− oocytes, while there was a clear change in gene expression in Dicer−/− oocytes compared to wild-type. Lack of obvious oocyte defects combined with lack of obvious gene expression changes in Dgcr8 mutant oocytes lead Suh et al.[9], to conclude that miRNAs do not function during mouse oogenesis. More recently, Freimer et al.[11], suggested that miRNAs are generated in oocytes but do not repress mRNA targets, because Argonaute 2 (Ago2) in GV oocytes is present in an alternate non-functional form[11]. These data lead Freimer et al.[11], to propose that miRNAs may be generated in oocytes but need a functional Ago2 protein to execute target repression. The function of miRNAs, if any, thus remains unclear during oocyte development.

Studies from *C. elegans*, on genetic mutants with systemic loss of Argonaute-like genes 1 and 2 (ALG-1/2), or Drosha and Pasha, however, revealed germline defects. For example, ALG-1/2 mutants display reduced germline stem cells[12]; and Drosha and/or Pasha mutants display defects in oocyte meiotic maturation[13]. These studies were conducted on mutants where gene function was removed in the entire animal rather than just the germline. Thus, it is currently unclear whether ALG-1/2, Drosha, and Pasha regulate germline development autonomously or non-autonomously. This is a key point, as Dicer acts non-autonomously in the somatic gonad to regulate oocyte meiotic maturation in worms and has no oogenic phenotype when depleted from the germline[10]. Moreover, in *C. elegans*, and in mice, Dicer was shown to regulate oocyte meiotic maturation through endo-siRNAs rather than miRNAs[10,14]. Thus, the role of miRNAs during oocyte development requires deeper investigation. In *C. elegans*, a cassette of ~30 miRNAs were identified bound to the germline-expressed Argonaute, ALG-5[15]. However, whether these ALG-5 bound miRNAs function to regulate oogenesis remains untested.

To resolve these questions, we performed an integrated functional genomic analysis, and profiled *C. elegans* adult hermaphroditic germlines for all known miRNAs. miRNAs identified as germline-enriched in the genomic analysis were then assayed for an oocyte expression via in situ hybridization. Oocyte-expressed miRNAs were tested for function using two different genetic methods. Our results identify that a repertoire of at least 23 oocyte-expressed miRNAs regulate oogenesis. The remaining ~93% of miRNAs were not detected in wild-type germlines, suggesting that they are either not generated or generated in amounts below a reliable detection limit. Unexpectedly, we find that most of the oocyte-expressed miRNAs are generated through an unknown Drosha independent manner. Consistent with them being *drosha*-independent these miRNAs do not overlap with the population of miRNAs identified to be enriched with ALG-5. These observations suggest that the miRNA biogenesis pathway may not function as a linear pathway as envisioned (with DRSH-1 → DCR-1 → ALG-5) but instead the pathway genes may be redundant with other currently unknown genes, especially at the Drosha processing step.

## Results

**Genomic methods identify germline miRNAs**. We profiled for 307 mature miRNAs using the FirePlex® platform previously described for miRNA profiling in *C. elegans*[16,17]. The FirePlex® analysis was conducted on RNA extracted from 100 dissected germline each from wild-type and *drsh-1(ok369)* (here on referred to as *drosha*) mutant animals, in biological triplicate at 18 h past the L4 stage of development. We used this stage of development for analysis since *drosha* mutants exhibit severe oocyte meiotic maturation at and after 24 h past the L4 stage (Supplementary Fig. 1). However, at earlier stages of oocyte development (such as 16–18 h past the L4 stage), the germlines from *drosha(ok369)* mutants do not appear malformed, despite a loss of *drosha* activity (Supplementary Fig. 1). These stages of development, thus enable us to assay for miRNAs that are specifically dependent on *drosha* activity in the absence of any gross morphological defects. This analysis identified 29 mature miRNAs from dissected wild-type adult germline (Fig. 1a and Supplementary Data 1); the remaining ~93% of miRNAs were detected at the threshold limit of the probes, suggesting that either these were background levels or very low amounts in either the wild-type or the *drosha* mutant gonads (Supplementary Data 1 and Supplementary Fig. 2b, $n = 276/307$).

Of the 29 miRNAs that were robustly detected in wild-type gonads 11 were reduced in *drosha* mutants suggesting that these arise in a *drosha*-dependent manner (Fig. 1a, Supplementary Data 1 and Supplementary Fig. 3a–d). Eighteen miRNAs were, however, detected at comparable or higher levels in *drosha* mutant gonads relative to wild-type (Fig. 1a, Supplementary Data 1, and Supplementary Fig. 3f–j). This suggests that these miRNAs arise in a *drosha*-independent manner, or perdure in the absence of Drosha activity. We also identified a previously described Drosha independent miRNA: miR-62, in *drosha* mutant gonads. miR-62 is a mirtron, wherein the pre-miRNA precursor resides in the intronic region and is generated in a Drosha independent manner[18]. The identification of miR-62 as *drosha*- independent species in this analysis is consistent with previous findings. Although the mechanisms through which miR-62 is generated may be distinct than those of other *drosha*-independent miRNAs identified in this study. A total of 19 miRNAs were identified in this analysis to be *drosha*-independent (Fig. 1a, Supplementary Data 1, and Supplementary Fig. 3e–j).

To independently assess the dependence of the miRNAs identified in the FirePlex® analysis on *drosha* activity, we performed TaqMan™ analysis using probes to mature miRNAs on wild-type and *drosha* mutant animals (Fig. 1b, c). The TaqMan™ assay was performed on RNA extracted (Methods) from wild-type animals, miRNA mutant animals (to check specificity of the TaqMan™ probe) and on *drosha(ok369)* animals at 18 h past the L4 stage of development (Supplementary Fig. 1). The *drosha*-dependent miRNAs tested were reduced in *drosha (ok369)* mutants relative to wild-type (Fig. 1b), while the *drosha*-independent miRNAs tested accumulate in the *drosha* mutants

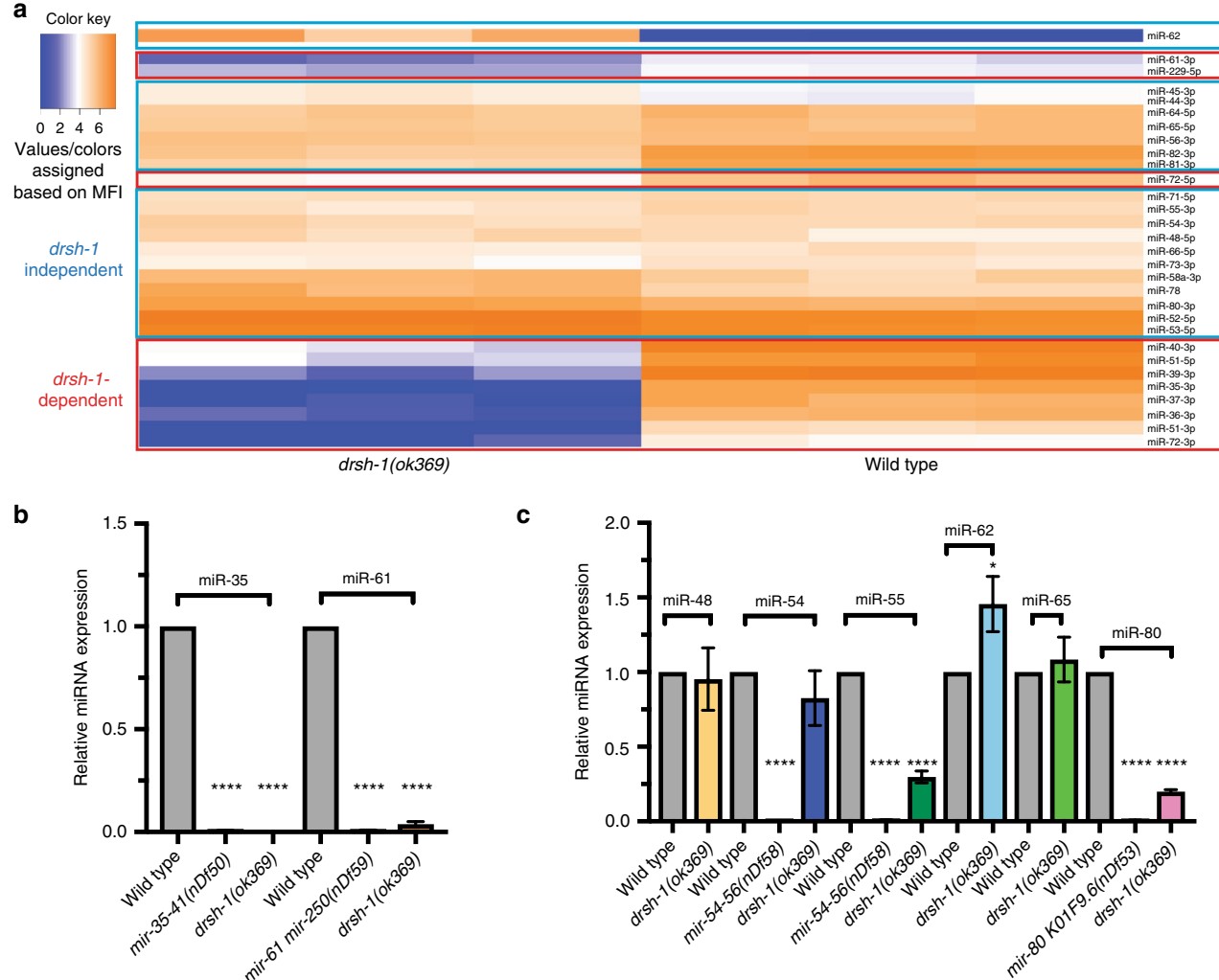

**Fig. 1** Germline-expressed miRNAs are largely *drosha*-independent. **a** FirePlex® assay-based heat map analysis of miRNAs on dissected wild-type and *drosha(ok369)* germlines at 20 °C. Values/colors reflect the mean fluorescent intensity (MFI). Deep blue = low miRNA accumulation. Bright orange = high miRNA accumulation. Red boxes: miRNAs that are reduced in *drosha(ok369)* germlines termed *drosha*-dependent. Blue boxes: miRNAs that accumulate in *drosha(ok369)* were termed *drosha*-independent. One-hundred dissected gonads were assayed for each genotype in three biological replicates. **b, c** TaqMan™ analysis on wild-type, *drosha(ok369)* and each of the miRNA mutant genotypes at 18 h past L4. *drosha*-dependent miRNAs do not accumulate in *drsh-1* mutants (**b**), while *drosha*-independent miRNAs are still expressed (**c**). Each TaqMan™ assay was performed three times on 100 whole worms each time for each genotype. Statistical significance was calculated by one-way ANOVA with the Bonferroni's correction. ****$P < 0.0001$. ± SD

(Fig. 1c), suggesting that these miRNAs are likely generated independent of *drosha* activity even outside of the germline. Altogether, these two data sets reveal that a large proportion (~93%) of miRNAs are either expressed at very low abundance or not expressed in wild-type gonads. Of the 7% that are expressed at detectable levels, over half arise in a *drosha*-independent manner, suggesting that redundant mechanisms, currently unknown, may process pri-miRNAs in the germline.

**Oocyte-expressed miRNAs accumulate in four spatial domains**. An adult hermaphroditic germline consists of germ cells at different chromosomal stages of development, ranging from germ-line stem cells to maturing and ovulating oocytes, as well as mature sperm[19]. To map the distinct regions of the germline to which the germline miRNAs localize, we used a locked nucleic acid (LNA) based in situ hybridization method and assayed dissected adult hermaphroditic germlines (Methods). The in situ hybridization method was performed on all 29 miRNAs identified by FirePlex® analysis (Supplementary Fig. 4). Specific miRNA

mutants were used as negative controls to detect probe specificity. Four miRNAs that were not detected as germline-enriched were also assayed to assess the specificity of the in situ method (Supplementary Data 1 and Supplementary Figs. 2 and 4). The U6 small nucleolar RNA was used as a positive control; U6 is expressed in all germ cell nuclei (including sperm, Fig. 2a, arrowhead) as well as the somatic sheath cells (Fig. 2a, arrow), and distal tip cell of the somatic gonad. Overall, 23 of the 29 miRNAs assayed displayed strong positive signal in the germline (Supplementary Fig. 4) and did not display any signal in sperm (Fig. 2 and Supplementary Figs. 5–10), somatic sheath cells or distal tip cell (Fig. 2 and Supplementary Figs. 5–10); six miRNA probes showed negligible signal in the germline (Supplementary Fig. 4). The last could be because these are expressed at very low levels and may be below the detection limit of the colorimetric method employed in the in situ hybridization method.

Twenty-three miRNAs displayed positive expression in overlapping as well as distinct regions of the germline. We used positive expression of the miRNA probes in different regions of the germline

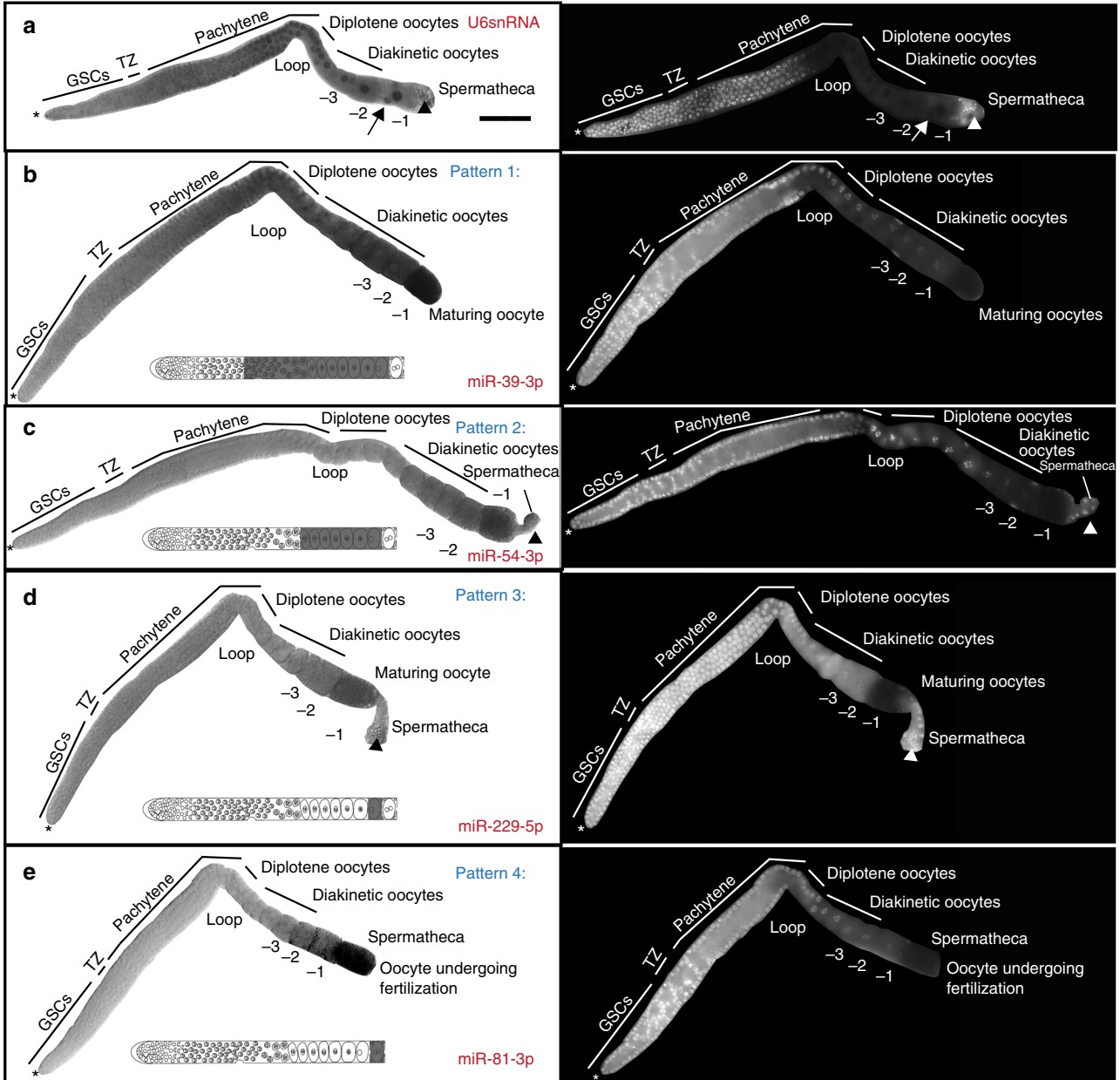

**Fig. 2** Oogenic miRNAs localize in four spatial patterns in the germline. Dissected germlines, oriented left to right, with oocytes on the right. Asterisk marks the distal tip cell. Bright-field (left) images of germlines. Dark staining is indicative of positive probe expression. DNA (white, stained with DAPI, right panel) staining labels distinct germ cell nuclear stages. The schematic in each panel depicts miRNA localization, gray labeling highlights positive probe expression. **a** U6 small nucleolar RNA, expresses in all germ cell nuclei (arrowhead marks sperm) and the somatic gonad nuclei (arrow marks somatic sheath cells). **b** Pattern 1 miRNA, miR-39-3p, localizes to the cytoplasm of late pachytene cells, diplotene oocytes, diakinetic oocytes, and maturing oocytes, but not in sperm cells or somatic sheath cells. **c** Pattern 2 miRNA, miR-54-3p, localizes to cytoplasm of diplotene oocytes, diakinetic oocytes and maturing oocytes. **d** Pattern 3 miRNA, miR-229-5p localizes to cytoplasm of the maturing oocyte. Arrowhead marks sperm, which are negative for the miRNA. **e** Pattern 4 miRNA, miR-81-3p localizes to the cytoplasm of an oocyte undergoing fertilization in the spermatheca. None of the miRNAs positively localized to the somatic gonadal cells or sperm cells. Each experiment was performed three times, 75–100 gonads were dissected for each replicate and ~20–30 germlines were imaged for each replicate. Scale bar: 20 μM

to define four overlapping spatial patterns. None of the miRNAs tested were localized to the germline stem cell population. Thus, the repertoire of germline-enriched miRNAs identified in this study are unlikely to regulate stem cell function. In addition, the 23 miRNAs displayed a positive signal in meiosis 1 oogenic cells, but not in mature sperm. miRNAs that localize to late pachytene cells, diplotene, diakinetic and maturing oocytes were defined as belonging to the same pattern, which we termed Pattern 1 (Fig. 2b and Supplementary Fig. 5). Seven miRNAs including miR-61 and

members of the miR-35 family belong to this pattern (Fig. 2b and Supplementary Fig. 5). Pattern 1 miRNAs were all *drosha*-dependent (Fig. 1, Supplementary Data 1, and Supplementary Fig. 3). miR-35 expression was absent in germlines of worms homozygous for a deficiency in the *mir-35–41* family (*nDF50*), confirming the specificity of the RNA probe as well as the in situ hybridization method (Supplementary Fig. 6).

miRNAs that localized to diakinetic and maturing oocytes were defined as Pattern 2 (Fig. 2c and Supplementary Fig. 7).

Seven miRNAs followed this expression pattern (Fig. 2c and Supplementary Fig. 7). Three of the seven Pattern 2 miRNAs were *drosha*-dependent (miR-51-5p, miR-72-3p, and miR-72-5p), and four were *drosha*-independent (miR-52-5p, miR-54-3p, miR-65-5p, and miR-78) in the genomic analysis (Fig. 1 and Supplementary Data 1). Thus, whether a miRNA is dependent on Drosha function or not does not distinguish its localization pattern in the germline.

Six miRNAs localized to maturing oocytes only and were defined as Pattern 3 miRNAs (Fig. 2d and Supplementary Fig. 8). One, of six, Pattern 3 miRNAs was *drosha*-dependent (miR-229-5p), and five were *drosha*-independent in the genomic analysis (miR-55-3p, miR-58a-3p, miR-71-5p, miR-73-3p, and miR-82-3p) (Fig. 1 and Supplementary Data 1).

Four miRNAs localized only to oocytes undergoing fertilization, and after ovulation in the spermatheca (miR-53-5p, miR-80-3p, and miR-81-3p) (Fig. 2e and Supplementary Fig. 9). We defined these as Pattern 4, and they were identified in the genomic analysis as *drosha*-independent (Fig. 1 and Supplementary Data 1).

Four miRNAs negative for germline expression via FirePlex® analysis were also negative in the in situ analysis (Supplementary Fig. 10).

Because all 23 miRNAs in the in situ analysis displayed a positive signal in oogenic germ cells but not in the mature sperm or the somatic gonad (specifically the distal tip cells and the somatic gonadal sheath cells), we refer to them as oocyte-expressed miRNAs. Of note, there was a trend in *drosha*-independent miRNAs expressing in germ cells as they progress towards oocytes. In a growing germline, oocytes start to individualize in the diplotene region, and transcription shuts down at the end of pachytene[20]. Thus, while the LNA probe used to map the expression of the miRNAs can detect mature, pre and primary miRNAs, the localization visualized above is likely not detecting active transcription and may be a true indicator of a processing-based accumulation of pre or mature miRNAs. Moreover, the expression patterns of these oocyte-expressed miRNAs suggest that they either function to regulate oocyte development or that they may be loaded into the embryo as maternal miRNAs for an embryonic function.

**Oocyte-expressed miRNAs regulate oogenesis.** To assess whether the oocyte-expressed miRNAs regulate oogenesis, we analyzed adult germlines from representative miRNA mutants of Patterns 1 through 4 using two different methods. (1) We obtained existing well-characterized mutant alleles[21] for representative miRNAs (Fig. 3 and Supplementary Fig. 11). (2) We performed RNAi mediated depletion of the pri-miRNAs to assess miRNA function for the same representative miRNAs obtained as genetic deletions. The RNAi analysis was performed to ensure that the effects, if any, on oogenesis were due to an RNA function of the miRNA being tested rather than because the genetic deletion was removing a long-range transcriptional enhancer element (for example), affecting a downstream gene unrelated to the miRNA being tested. The efficiency of RNAi mediated depletion was assayed by TaqMan™ based quantitative reverse transcription PCR (qRT-PCR) analysis of mature miRNAs from each of the RNAi treatments (Supplementary Fig. 12d). As a negative control Luciferase RNAi (Fig. 4a) or no dsRNA treatment was used to assess the effects of a control dsRNA (that should not target any mRNA) and starvation (which occurs on soaking), respectively (Supplementary Fig. 12a, b). As a positive control *mpk-1* RNAi (Supplementary Fig. 12c) was used and revealed previously characterized meiotic progression and large oocyte defects[22].

Each of the miRNA mutant analyzed here was homozygous viable as shown in the study that generated the alleles[21], thus even if there were to be defects during oocyte development, we surmised they may be subtle and only visible on phenotypic analysis of dissected adult oogenic germlines. Thus, we analyzed dissected germlines from wild-type and mutant animals at two stages of development, 18 and 24 h past the L4 stage, to assess meiotic progression, oocyte development and meiotic maturation (which starts at 20–22 h past the L4 stage of development and occurs every 20 min). We also assayed brood size as an indicator of productive oogenesis, and hatching of the F1 embryos, produced by each of miRNA mutants, as an indicator of normal embryonic development. We surmised that if there were defects in fertility of the miRNA mutant animals, then these would be interpreted as a lower number of progenies. The RNAi analysis was performed on L4 animals for 24 h using soaking RNAi (Methods).

Adult germlines from each of the miRNA mutant animals were analyzed upon dissection and staining for DNA (DAPI), lamin (to mark the nuclear membrane, using anti-LMN-1 antibody) and nucleolus (using anti-NOP-1 antibody, Methods). RNAi analysis against the pri-miRNAs was conducted with live imaging on animals carrying germline-expressed membrane tagged GFP reporter to visualize germ cell plasma membranes, and Histone 2B tagged with mCherry to visualize the germ cell nuclei (Methods). As detailed below, our approach revealed germline defects using both the deletion mutant analysis as well as RNAi analysis for each of the Pattern 1 through 4 miRNAs in the spatial domain in which the miRNA was expressed.

From Pattern 1 miRNAs, we assayed the dissected germlines from the *nDf50* allele, that deletes *mir-35–41* and *nDf59* allele, which deletes *mir-61* and *mir-250*. We note here that miR-250 was not identified as an oocyte-expressed miRNA in our analysis. However, the characterized available allele removes the precursors for both miR-61 as well as miR-250. Dissected adult (24 h past L4) germlines from hermaphroditic animals' mutant for either *nDf50* and *nDf59* revealed pachytene progression defects via DAPI analysis used to visualize DNA morphology (Fig. 3b, c). In wild-type adult hermaphroditic germlines, pachytene stage cells transition to diplotene oocytes at the loop region. However, in the adult germlines from the mutant alleles, the transition from pachytene to diplotene was delayed, compare inset Fig. 3b (*mir-35–41* (*nDf50*)) and Fig. 3c (*mir-61mir-250*(*nDf59*)) to Fig. 3a (wild-type). This delay causes defects in meiotic I progression[22,23]. In addition, the diplotene and diakinetic oocytes were reduced in number in the mutant germlines relative to wild-type, as assessed by LMN-1 which marks each growing and arrested oocyte nucleus (Fig. 3). As in the deletion mutants, we observed pachytene progression defects as well as lower number of oocytes on whole germlines unperturbed from any dissection-based treatments from RNAi analysis for miR-35–41 and miR-61 (Fig. 4b, c, g, h).

To determine whether the observed germline defects resulted in reduced fertility or reduced embryonic viability, we assayed brood size and embryonic lethality from each of the miRNA mutants. Both *mir-35–41(nDf50)* and *mir-61mir-250(nDf59)* display reduced brood size. The reduction in brood size from the mutant hermaphroditic animals was by almost half that of wild-type animals, suggesting that the oocyte defects visualized in germlines from each of the mutant animals result in reduced fertility (Fig. 5a).

*mir-35–41(nDf50)* also displayed embryonic lethality, as shown previously[24,25] (Fig. 5b), while *mir-61mir-250(nDf59)* did not display any embryonic lethality (Fig. 5b). These data suggest the *mir-35–41* family regulates both oogenesis and embryogenesis, while *mir-61mir-250* regulates oogenesis. The observed germline

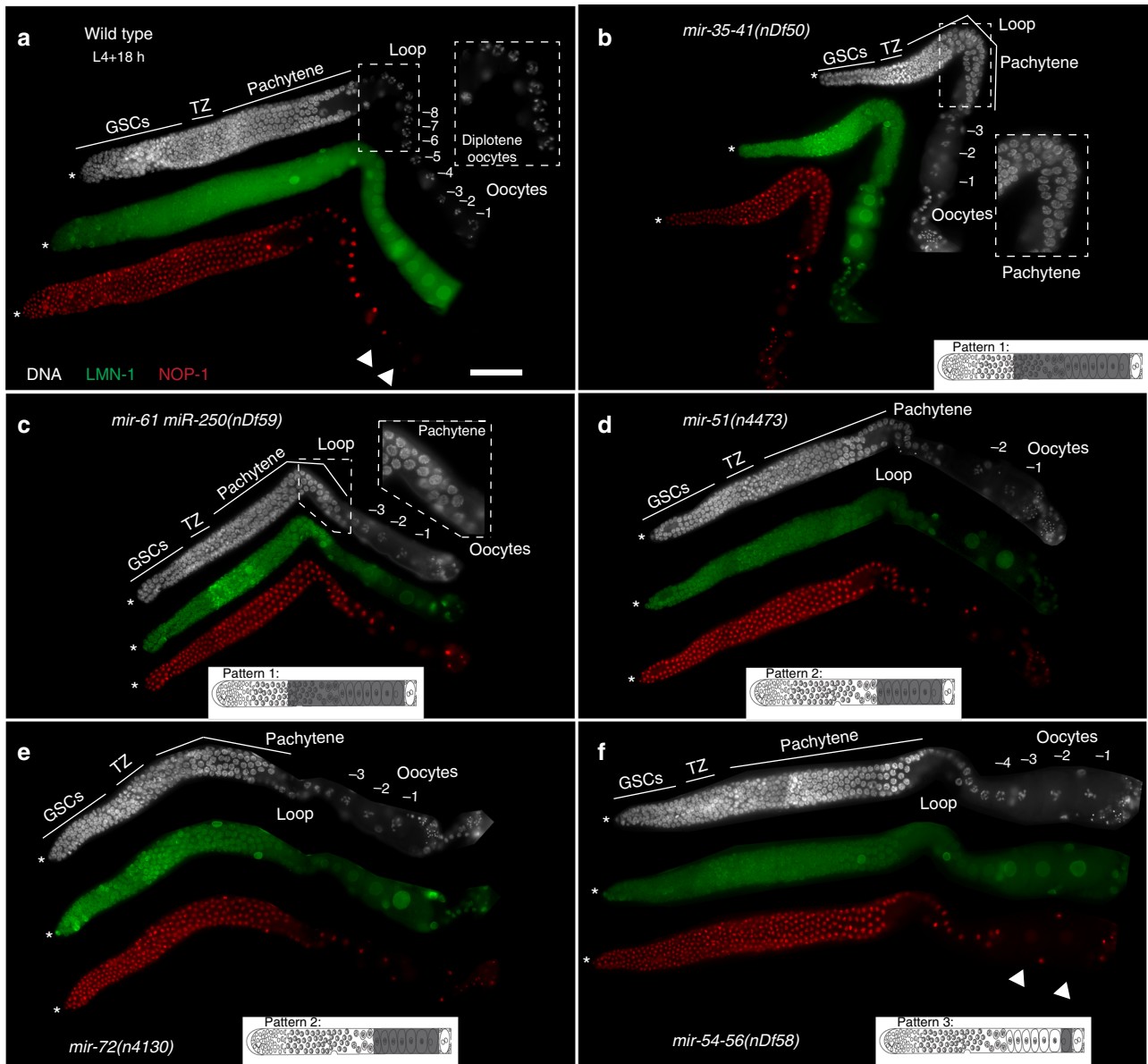

**Fig. 3** Oocyte-expressed miRNAs regulate meiotic progression, and oocyte development. Dissected germlines stained with DAPI to visualize nuclear morphology (DNA, white), NOP-1 (nucleolus, red), and LMN-1 (lamin, green) oriented from left to right with oocytes on the right. Asterisk marks the distal tip cell. The schematic in each panel is indicative of miRNA localization (in gray) in an adult germline. **a** Wild-type germlines exhibit 7–8 oocytes with no oogenic defects. **b** *mir-35–41(nDf50)*, mutant germlines display pachytene progression defects (inset shows pachytene cells in the loop region, where normally germ cells exit into diplotene in wild-type) and reduced oocyte number relative to wild-type at 24 h past L4 stage of development. **c** *mir-61 mir-250(nDf59)*, mutant adult germlines display delayed pachytene progression and reduced oocyte number. **d** *mir-51(n4473)*, mutant adult germlines display lower oocyte number compared to wild-type. **e** *mir-72(n4130)*, mutant adult germlines display reduced oocyte number compared to wild-type. **f** *mir-54–56 (nDf58)*, mutant adult germlines display lower oocyte number, with each oocyte also exhibiting abnormal shape relative to wild-type. NOP-1 is abrogated in the -2 oocyte (arrow) unlike wild-type germlines (**a**, arrow), where NOP-1 is visible in the -2 oocyte. Each experiment was performed three times, 50–75 gonads were dissected for each replicate from each genotype, and ~20–25 germlines were imaged per replicate. Scale bar: 20 μM

phenotypes correlate with the localization of miR-35–40 and miR-61 in the germline.

We assayed two candidate Pattern 2 miRNAs: miR-51 and miR-72 for germline functions using the available mutants *mir-51 (n4473)* and *mir-72(n4130)* and upon RNAi analysis against the pri-miRNAs for miR-51 and miR-72. We observed reduced number of oocytes in adult germlines at 24 h past L4 from *mir-51 (n4473)* and *mir-72(n4130)* mutant hermaphroditic animals as well in animals treated with RNAi for *mir-51* and *mir-72* (Figs. 3d, e and 4d, e, g), relative to wild-type germlines. RNAi analysis of each of the two miRNA genes also revealed non-linear

organization of diplotene oocytes. Because Pattern 2 miRNAs localize starting at diplotene oocytes in the hermaphroditic germline (Fig. 2), we surmise that the defect in diplotene organization may be due to the function of these miRNAs to regulate the transition of chromosomes from pachytene or diplotene of meiosis or they may function to organize the linear row of growing diplotene oocytes (Fig. 4h). In addition, the mutant hermaphroditic animals from each miRNA displayed a reduction in brood size (Fig. 5a). Once born the embryos from *mir-51(n4473)* and *mir-72(n4130)* mutant hermaphroditic animals were viable (Fig. 5b). These data suggest that Pattern 2

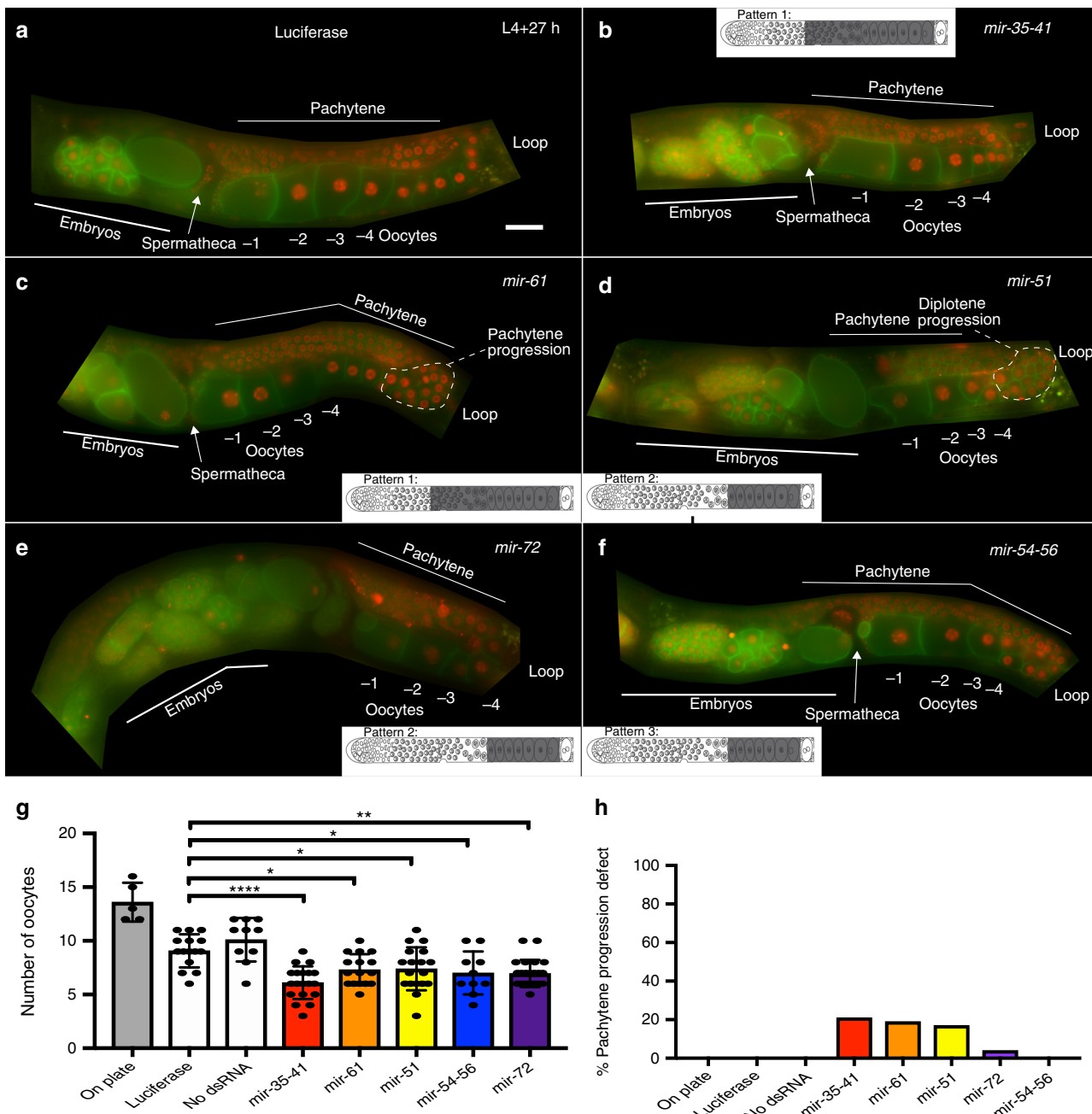

**Fig. 4** RNAi mediated depletion of oocyte-expressed miRNAs reveals defects in meiotic progression and oocyte development. The photograph displays live imaging of germlines from whole animals bearing membrane GFP (green) and Histone 2B mCherry (red). The animal is oriented in dorsal (top) ventral (bottom) polarity. The schematic in each panel depicts the pattern of accumulation, gray indicates regions of positive localization. **a** Luciferase RNAi does not produce oocyte defects. See also Supplementary Fig. 12 for negative controls. **b** *mir-35–41* RNAi results in a lower number of oocytes and delayed pachytene progression. **c** *mir-61* RNAi results in a lower number of oocytes and delayed pachytene progression. **d** *mir-51* RNAi causes oocyte disorganization and double row of diplotene oocytes, coupled with lower oocyte number. **e** *mir-72* RNAi results in a lower number of oocytes. **f** *mir-54–56* RNAi results in a lower number of oocytes. **g** Quantitation of oocyte number from each of the RNAi treatments. Each oocyte is numbered from diplotene stage (starting at the loop region). Each experiment was performed three times, ~20–25 animals assayed per replicate and statistical significance was calculated by one-way ANOVA with Bonferroni's correction. ****$P < 0.0001$, **$P < 0.001$, *$P < 0.01$. ±SD. Scale bar: 20 μM. **h** Percentage of animals displaying pachytene progression defects quantified for each RNAi treatment taking all three replicates into consideration

miRNAs regulate meiotic progression and oocyte development, but not embryonic development.

From Pattern 3 miRNAs we assayed *mir-54–56(nDf58)* mutant hermaphroditic animals. *nDf58* deletes the pre-miRNA region of miR-54, 55, and 56. RNAi analysis was also performed on this pre-miRNA region, and thus depleted all three miRNAs: miR-54, 55, and 56. Of the three miRNAs in this cluster, in situ hybridization analysis for miR-54 displayed Pattern 3 like expression (Fig. 2c), miR-55 displayed Pattern 4 like expression (Supplementary Fig. 8). miR-56 was not identified in the genomic analysis to be germline expressed (Supplementary Data 1). We observed that adult germlines at 24 h past the L4 stage of analysis from *mir-54–56(nDf58)* mutant animals as well as upon RNAi treatment of miR-54–56 displayed reduced number of oocytes

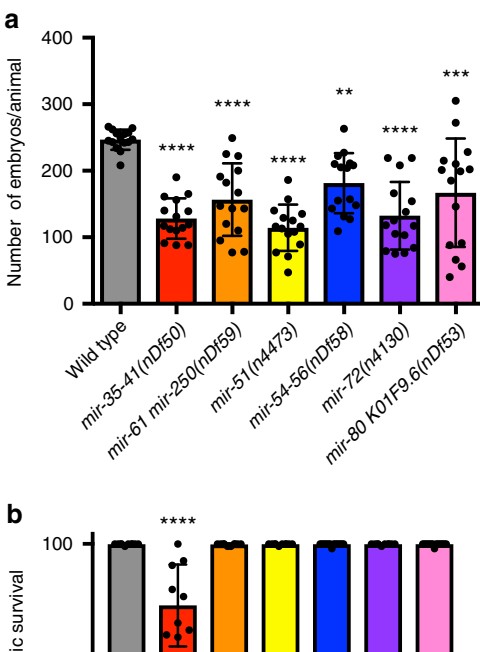

**Fig. 5** Oocyte-expressed miRNAs regulate fertility. **a** Bar graphs represent number of embryos laid per animal over the course of its life time. Wild-type animals generate ~250 embryos at 20 °C, each of the miRNA mutants generated 100–150 embryos. Each experiment was repeated three times, and each time five animals were counted for their number of progenies. Statistical significance was calculated with one-way ANOVA with the Bonferroni's correction. ****$P < 0.0001$, **$P < 0.01$. ± SD. **b** Bar graphs represent the percent of embryos that hatch and develop into larvae. Embryos from *mir-35–41* mutants display reduced embryonic survival. Each experiment was repeated three times, and each time five animals were counted for any embryonic lethality of their progeny. All the embryos deposited on the plate by each mutant were assayed for development to larval stages. Lack of development to larval stage was scored as embryonic lethality. Statistical significance was calculated by one-way ANOVA with the Bonferroni's correction. ****$P < 0.0001$. ± SD

(Figs. 3f and 4f, g). These data suggest that *mir-54–56* regulates oocyte development. Based on the expression pattern of Pattern 3 and 4 miRNAs, we also wondered whether the mutant hermaphroditic animals display any meiotic maturation defects. To assess whether the mutant animals displayed a subtle meiotic maturation defect (since we did not observe any gross defects in meiotic maturation, such as endomitotic oocytes), we used NOP-1 the nucleolar protein which breaks down in preparation of meiotic maturation in the −1 oocyte. Precocious loss of NOP-1 would be indicative or early meiotic maturation defects as shown previously[26]. Consistent with this hypothesis, we found that NOP-1 is lost precociously in mutant germlines (Fig. 3f, arrowhead). The precocious loss of nucleolar marker in *nDf58* germlines suggests early (and thus abnormal) onset of oocyte meiotic maturation. In addition, the mutant displayed lower

brood size, although the embryos' born were viable. These data are indicative of lower fertility and oogenic defects (Fig. 5).

For Pattern 4 miRNAs, we assayed animals carrying the *nDf53* deletion which removes miR-80 as well as K01F9.6. Pattern 4 miRNAs localize to oocytes undergoing fertilization upon ovulation in the spermatheca (Fig. 2e and Supplementary Fig. 9). Adult (24 h past L4) hermaphroditic germlines from the *nDf53* mutants exhibit normal oocyte development and number (Supplementary Fig. 11). However, the mutant worms displayed a significantly reduced brood size (Fig. 5a). Thus, the *nDf53* mutant animals displayed (a) no obvious oocyte defect and (b) lower brood size. Superimposing these observations with the in situ expression pattern for miR-80 which was restricted only to the oocyte undergoing fertilization, and was not apparent either in the sperm or developing oocytes, we conclude that miR-80 may regulate oocyte to embryo transition, and thus loss of *mir-80* may result in lower brood size.

Altogether these data reveal that oocyte-expressed miRNAs localize to distinct regions of the germline defined by the chromosomal state of the germ cells (for example, pachytene stage, or diakinetic stage). Functional analysis demonstrates that the oocyte-expressed miRNAs regulate specific oocyte progression and developmental phenotypes in their region of expression. The oogenic defects result in reduced fertility of the animals, as assessed by brood size analysis.

## Discussion

We identify a repertoire of oocyte-expressed miRNAs that regulate oogenesis. Prior to our work, the role of miRNAs in oogenesis in worms and mice was debated, and several conflicting models existed. One model suggests that miRNAs do not function during oogenesis since germline specific depletion of miRNA processing enzymes had little effect on oogenesis, despite a reduction in miRNAs (in mice). A second model suggests that miRNAs are generated in oocytes but function during embryogenesis. We find that neither of these models applies to miRNA function in *C. elegans* oogenesis. Instead, a specific repertoire of miRNAs is expressed and regulates key developmental events during meiosis I of oogenesis, such as pachytene progression, oocyte development, maturation, and the transition of an oocyte to an embryo.

In *C. elegans*, systemic loss of *drosha*, *pasha* and *dicer* results in oocyte meiotic maturation defects[27]. However, mosaic analysis with depletion of *dicer* specifically from the germline did not result in oocyte meiotic maturation defects[10], in addition in the *dicer* mutants 26G endo-siRNAs were abrogated relative to the miRNAs, the latter seemingly perdure. These observations led to open questions in the field as to whether miRNAs were expressed and functional during *C. elegans* oogenesis. Our study demonstrates that (a) miRNAs are expressed and (b) the functional importance of miRNAs during oogenesis. We identified at least 23 miRNAs expressed during oogenesis. The remaining ~93% of miRNAs were not detected above background in this analysis; these are either produced at very low levels or not produced in the germline. 17, of the 23 miRNAs, assayed using six different alleles or RNAi treatments yielded clear defects in oocyte development and brood size (Figs. 3–5) suggesting that these are not only expressed in oocytes but function to regulate oogenic development. The miRNA mutants analyzed in this study were previously generated and broadly characterized for phenotypes[21]. The previous study assayed for egg retention in the uterus, as a measure of embryo production, and did not directly assay for germline defects. The analysis of germline development in this study was designed to assay for specific oogenic defects. Because genomic deletions can often delete enhancer elements in

noncoding regions, we also assayed RNAi mediated depletion and found that the phenotypes observed are miRNA dependent. Although our study relied on systemic knockdown of miRNA function, we expect the observed phenotypes to arise due to germline functions of miRNAs for two reasons: none of the 23 miRNAs were expressed in the somatic gonadal sheath cells, distal tip cell, or sperm (Fig. 3 and Supplementary Figs. 4–5 and 7–9); the germline phenotypes observed for each miRNA mapped to its corresponding expression in the germline (Figs. 3 and 4).

Comparing the analysis in this study to two previous studies that identified germline-expressed miRNAs in *C. elegans*, we find that McEwen et al.[28], identified 13 germline-enriched miRNAs via Next generation RNA-Sequencing analysis; only seven miR-NAs from this analysis (miR-35, miR-36, miR-37, miR-38, miR-39, miR-40, and miR-56) overlap with our data set. The remaining five were identified as negligibly produced in wild-type germlines (Supplementary Fig. 2 and Supplementary Data 1) from the FirePlex® assay in this study and may reflect a difference in the analysis between the two studies. McEwen et al., performed RNA-sequencing analysis on whole animals and identified the germline miRNAs by differential expression between *dicer* mutants, wild-type and germline defective mutants, while we directly assayed wild-type dissected adult germlines using the particle based FirePlex® analysis. Although our analysis did not detect miR-250 as germline-expressed as shown in McEwen et al., we did perform in situ hybridization on miR-250. We did not observe any signal over background on dissected germlines using the miR-250 LNA probe. Thus, we conclude that miR-250 may either not be expressed in adult hermaphroditic germlines, or may be expressed at levels below the detection limit of our in situ method.

Brown et al.[15], identified miRNAs that are enriched in an immuno-precipitate with the germline-expressed Argonaute ALG-5. Comparison of the Brown et al., work with our analysis revealed that 9 out of 10 *drosha*-dependent miRNAs identified in this study are ALG-5 enriched, while none of the *drosha*-independent species were ALG-5 enriched (Supplementary Fig. 13). Thus, the *drosha*-independent miRNAs may use a distinct Argonaute for their function. Brown et al., also identified miR-NAs with the ALG-5 immuno-precipitate which we did not identify as germline expressed in our analysis (Supplementary Fig. 13). This could be because ALG-5 is more broadly expressed than just the germline, for example during embryogenesis, or it could be that immunoprecipitation enriches for specific populations, which we did not conduct. In addition to these two studies, Martinez et al.[29], generated promoter based transgenic lines with the promoters from each of the miRNAs and identified miR-246 as being expressed in the somatic gonad. From our dissected gonadal analysis, that retains the somatic gonad components of the sheath cells, and the distal tip cell, we did not identify miR-246 as being expressed in the gonad using FirePlex® analysis. It is likely that miR-246 is expressed more abundantly in the spermathecal-uterine valve and the uterine tissue of the somatic gonad rather than the distal somatic sheath cells and thus our analysis did not positively detect it. Altogether, our study provides a comprehensive analysis of the germline-expressed and functional miRNAs.

Of the 29 miRNAs expressed in the germline identified from the FirePlex® analysis, over half were also positive in *drosha (ok369)* mutant germlines, suggesting that these accumulate in the absence of *drosha* function. Our use of TaqMan™ analysis on *drosha* mutants, at 18 h past the L4 stage of development, independently validates that the *drosha*-independent miRNAs bypass a requirement for Drosha function in vivo (even outside of germline, since we used whole animals for this analysis) (Fig. 1). The *drosha(ok369)* mutant germlines at 18 h past L4 stage of development however reveal very low expression of *drosha* mRNA relative to wild-type (Supplementary Fig. 14), although at this time we cannot rule out that the maternal Drosha protein perdures in this mutant. Alternatively, it is likely, that the *drosha*-independent miRNAs may arise because pre-miRNAs once generated early in the life of the animal perdure into adulthood, finally it is possible that the *drosha*-independent miRNAs are generated through other redundant mechanisms. We tested the requirement for Dicer in processing the *drosha*-independent miRNAs via TaqMan™ analysis, however, as we had found in our previous analysis using Next generation sequencing[10], we found that miRNAs perdured in the absence of Dicer (Supplementary Fig. 15). These miRNAs are thus generated in a potential novel or redundant mechanism in vivo.

If half or more of the germline-expressed miRNAs arise in the absence of *drosha* function, what other enzymes and pathways might promote their biogenesis? Certainly, mirtrons have been shown to be generated independently of Drosha function[18,30], and miR-62, which was identified in our analysis as independent of Drosha activity is a mirtron[18]. In this case, splicing factors function to generate the pre-miRNA in a Drosha independent manner. However, perusal of the genomic locations of the oocyte-expressed miRNAs does not identify any additional *drosha*-independent miRNAs to be mirtrons (Supplementary Fig. 16). It is very likely that several redundant processing mechanisms exist that may generate the *drosha*-independent miRNAs. For example, RNAse III enzymes may perform this function, such as the Ago2, which can generate miRNAs independent of Dicer in vertebrates[31]. Alternatively, many RNA binding proteins, and the exosome proteins have been recently identified to promote miRNA processing[30,32]. As both splicing factors and Drosha are thought to function co-transcriptionally, some of these factors may act in parallel to Drosha or be able to compensate for the loss of Drosha to promote miRNA biogenesis. At this point, such factors remain unknown, but our identification of miRNAs that arise in the absence of *drosha* function provides an essential molecular handle to screen for these factors.

In summary, we identify the functional miRNA repertoire regulating oogenesis during meiosis I in *C. elegans*. Previous work in *C. elegans* showed that loss of Dicer from the germline did not result in any oocyte defects, however, in this case, the miRNAs perdured thus commenting on a direct function of miRNAs in this context is difficult. In mammals, miRNAs are thought to be dispensable for regulating oocyte development[9,33] and the post-transcriptional gene regulation seems to be funneled through endo-siRNAs to regulate oocyte development[14]. Our work suggests that not only does a small repertoire of miRNAs function to regulate oocyte development, but that in addition to canonical pathways for miRNA production, there may be non-canonical and redundant mechanisms that generate functional miRNAs driving oocyte development. We conclude that post-transcriptional gene regulation in oocytes also occurs through specific oocyte-expressed miRNAs, and miRNA function in regulating oocyte development may not be completely dispensable even in the mammalian context.

## Methods

**Experimental model and subject details**. Standard *C. elegans* culture conditions at 20 °C were used unless noted. Following alleles were used in this study. Linkage group I: *drsh-1(ok369)/hT2G*. Linkage group II: *alg-2(ok304)*, *mir-35–41(nDf50)*, *mir-72(n4130)*. Linkage group III: *mir-80(nDf53)*, *dcr-1(ok247)/hT2G* Linkage group IV: *mir-51(n4473)*; *mir-54–56(nDf58)*; Linkage group V: *mir-61mir-250 (nDf59)*. Transgenic line ltIs38[pAA1; pie-1::GFP::PH(PLC1delta1) + unc-119(+)] III; ltIs37 [P-pie-1::mCherry::his-58 (pAA64) + unc-119(+)] IV.

**FirePlex® experimental details**. FirePlex® is a hybridization-based detection method, it provides a better signal-to-noise ratio and is sensitive to the low amount

of starting material[34,35]. Total RNA (from three biological replicates) was extracted using the Nucleospin XS reagent from Macherey-Nagel. Specifically, the total RNA was extracted in the miRNA homogenate additive consisting of 2 M Sodium Acetate, pH. 4, and extracted by phenol chloroform. Small RNAs were extracted by size fractionation from each genotype and FirePlex® analysis performed as described (Methods). The average mean fluorescence value (MFI) of each miRNA sample analyzed was exported and raw data was used for analysis (Supplementary Data 1). Seven probes were used for each miRNA in seven distinct wells. These generated built-in redundancies and accounted for any well to well variation in signal. Data from the seven wells was then compressed into a merged-raw data set such that each of the 309 miRNAs had single data point for comparison, comprised of the averaged data. The average raw data from each of the 309 miRNAs was subtracted from average data from two negative controls (water). This generated the background-subtracted value for each miRNA. Positive controls cel-miR-71-5p, cel-miR-56-3p and cel-miR-52-5p were spiked into distinct lanes, and hybridized with corresponding probes to normalize for amounts. Samples were processed using the Multiplex Circulating miRNA Assay (Abcam) as per the published protocol with minor modifications to accommodate the use of multiple panels. FirePlex® Particles for each of five custom panels was added to a single well of a 96 well plate, followed by total RNA. These were hybridized at 37 °C for 60 min with shaking, followed by two rinses and the Labeling Buffer. The Labeling Buffer reaction was conducted at room temperature for 60 min with shaking. This resulted in the ligation of universal adapters to the ends of the hybridized miRNAs. The miRNAs were then eluted from the particles and were amplified by PCR with biotinylated primers that hybridize to the universal adapters. Following PCR amplification, miRNAs are re-hybridized to the probes. After additional rinses Reporter Mix containing fluorescently labeled biotinylated probes were hybridized to the particles. The samples were scanned on an EMD Millipore Guava 6HT flow cytometer for analysis.

**FirePlex® data analysis**. The flow cytometer output was analyzed with the Fire-Plex® Analysis Workbench software. The average mean fluorescence value (MFI) of all of the particles for a given target in a given well were exported and considered raw signals for each well. Three of the seven overlapping probes were then used to normalize the data between the five panels for a given sample, to account for any well-to-well variation. This data was then compressed into a merged-raw data set such that each of the 309 targets per sample had a single data point (for the seven overlapping targets, the average was used). The average of the signals from two negative control wells (where water had been added instead of sample) was subtracted from each sample to generate background-subtracted values. The genorm algorithm was used to identify cel-miR-71-5p, cel-miR-56-3p, and cel-miR-52-5p as optimal targets for normalization. The limit of detection of a probe is calculated from the negative samples, using a combination of two factors. Let $S_{ww}$ be the well-to-well standard deviation of the merged signals, and $S_{pp}$ be the mean of the particle-particle standard errors. Then the limit of detection is calculated as: $\sqrt{(3S_{ww})^2 + (4S_{pp})^2}$. Differential expression was determined by a two-tailed Student's T-test with significance calculates using an adjusted P-value, to correct for multiple comparisons across miRNAs. Multiple comparison correction was applied with Bonferroni's correction. Significance was set at $P < 0.05$.

**TaqMan™ analysis for detection of mature miRNAs**. RNA was collected from indicated strains as well as from RNAi treatments using TRIZOL™ Reagent (Invitrogen), or Nucleospin XS reagent from Macherey-Nagel. The reagent and method of extraction was kept internally consistent through each biological replicate. Three biological replicates were performed for each genotype for each experiment, each experiment had four technical replicates. Total RNA was purified using the miRNAeasy Mini Kit (Qiagen). Complementary DNA was generated using custom TaqMan™ assay primers (Applied Biosystems) from 10 ng of total RNA. qRT-PCR analysis was performed with specific TaqMan™ assays for each miRNA using manufacturer protocols (Applied Biosystems). Expression levels were standardized to a U6snRNA positive control and then to wild-type using the $\Delta\Delta C_t$ method of standardization.

**Gonad dissection and analysis**. Dissections of adult worms at indicated stage of development to obtain intact gonads were performed[23,36], under 5 min (immediately after adding levamisole). The dissected gonads were then fixed in 3% paraformaldehyde for ten min, followed by a post-fix in 100% methanol at −20 °C overnight. The fixed and permeabilized gonads were then washed three times with 1X PBS containing 0.1% Tween-20 and processed for immunofluorescence staining as described[22,23,36].

**Antibodies**. The following antibodies were used for immunofluorescence analysis: anti-CeLMN used at 1:200 (gift from Dr. Kelly Liu, Columbia University). anti-NOP1 (Clone 28F2, ThermoFisher) used at 1:500. Secondary antibodies were donkey anti-mouse Alexa Fluor 594, goat anti-guinea pig Alexa Fluor 488 (all obtained from Molecular Probes, Eugene, OR). All secondary antibodies were used at 1:400. To visualize DNA, the fixed and stained samples were incubated with 100 ng/mL 4′,6′-diamidino-2-phenylindole hydrochloride (DAPI) in 1X PBS with 0.1% Tween-20 for 5 min at room temperature.

**In situ hybridizations**. Gonads were dissected[37] from indicated genotypes as described above, and fixed in 3% paraformaldehyde/0.25% glutaraldehyde/0.1 M $K_2HPO_4$ for 2 h at room temperature. Fixed gonads were washed three times in 1x PBS with 0.1% Tween-20 and stored overnight in Methanol at −20 °C. The gonads were then permeabilized with Proteinase K for 20 min and re-fixed in 3% paraformaldehyde/0.25% glutaraldehyde followed by a wash with 0.2% glycine and 0.13 M 1-methylimidazole, 300 mM NaCl. Upon treatment with 1-ethyl-3-(3-dimethylaminopropyl) carbodiimide (EDC) for 2 h at 25 °C, the gonads were hybridized with the miRNA probe (Exiqon A/S) for 16 h at 30 °C below the probes RNA melting temperature. The DIG labeled probes were developed using anti-DIG alkaline phosphatase antibody followed with a colorimetric reaction until signal was visualized under a dissection microscope. Gonads were stained with DAPI, mounted on slides, and visualized the following day using Zeiss Axiovision.

**Hairpin chain reaction based in situ mRNA hybridization**. drosha mRNA FISH was performed using hairpin chain reaction on dissected adult gonads at 24 h past the L4 stage[38–41]. The gonads were then fixed in 3% paraformaldehyde, 0.25% glutaraldehyde solution at room temperature for 2 h. The probes were obtained from Molecular Technologies. Inc (Berkley, CA) and manufacturer's instructions followed.

**RNAi by soaking**. Primary miRNAs were generated by PCR and cloned into the pCR II-TOPO vector for the indicated miRNAs. The vector was then transformed into Top10 competent cells and positive clones were verified by sequencing. dsRNAs were generated from each miRNA verified clone by HiScribe T7/Sp6 enzymes in vitro transcription reaction (New England Biolabs). The dsRNAs was purified using Phenol Chloroform extraction. 10 L4 animals were soaked for 24 h at 20 °C in 1 μg/μL of RNA and then recovered on plates at 25 °C for 3 h followed by assaying for oocyte development defects. Three replicates were completed for each miRNA.

**Brood size/embryonic lethality analysis**. Single L4 worms were placed on individual plates. Five worms were examined for each genotype indicated, and a total of three replicates were performed. Every 24 h, the parent worm was moved to a fresh plate and the total number of embryos was counted on the original plate. This process was repeated over several days until worms stopped producing progeny. The total number of progeny was calculated for each animal and then averaged between replicates. For embryonic lethality the number of dead larvae were counted for each animal and averaged between replicates. Differences in brood size or embryonic lethality were determined by one-way ANOVA with a Bonferroni's correction with significance defined as $P < 0.05$.

**Quantification and statistical analysis**. Statistics were run using Prism 7. Statistical details of experiments can be found in the Figure legends. Significance was defined as $P < 0.05$ for each statistical analysis used.

## Data availability

The data that support the findings of this study are available from the corresponding author upon request. A reporting summary for this Article is available as a Supplementary Information file.

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

## Acknowledgements

We thank Dr. James Skeath for critical reading of the manuscript and helpful suggestions. We thank members of the Arur Lab for critical discussions throughout the study. This work is funded by NIH GM98200. Work in the Arur Lab is supported by CPRIT RP160023, ACS RSG-014-044 DDC, Anna Fuller Foundation Award. S.A. is an Andrew Sabin Family Foundation Fellow at the University of Texas MD Anderson Cancer Center. This work was presented in part at the 2017 International *C. elegans* Meeting and the 2018 Non-coding RNA Keystone meeting in a talk format by A.L.M and we thank the community of scientists at these meetings for all their excellent feedback that helped strengthen the study. Strains were provided by the CGC, which is funded by the NIH Office of Research Infrastructure Programs (P40 OD010440). We thank Dave Reiner for the Luciferase RNAi plasmid.

## Author contributions

A.L.M. performed experiments. M.R.T., E.A. and G.T. performed the FirePlex® analysis and designed the chips for specificity testing. A.L.M. and S.A. designed the study, analyzed data, and wrote the manuscript.

## Additional information

**Competing interests:** A.L.M. and S.A. have no competing interests. M.R.T., E.A. and G.T. are employees of Abcam Inc.

