## [Peer Review File · Nature Communications]

Reviewers' comments:

Reviewer #1 (Remarks to the Author):

In this manuscript, Minogue et al., identify a set of microRNAs that are expressed in the *C. elegans* hermaphrodite gonad arm, specifically in the germ line. This work is mostly descriptive, focusing on the identification and possible functions of microRNAs in the process of oogenesis and oocyte maturation. The authors use the FirePlex microRNA assay that allows the researchers to analyze RNA samples for expression of 307 candidate microRNAs found in the miRbase database. Additional work used in situ hybridization and qPCR Taqman assays to validate and further define the spatial expression patterns. Lastly, preliminary functional analysis of oogenic microRNAs was performed using available mutants and a novel approach of using RNAi directed against the pre-microRNA. Results from functional analysis suggest a role for these oogenic microRNAs in the control of meiotic progression in the hermaphrodite germ line. These results advance our knowledge of the identification and functions of microRNAs in oocyte formation and maturation. This is an important contribution that provides a foundation for future work to delineate specific functions of individual microRNAs. There are a number of points that need to be addressed that would strengthen this manuscript.

Major points

1) A key point that the authors make is that the role of microRNAs in oogenesis is a central unknown due to conflicting results in different organisms, specifically data in mice that indicates that microRNAs are not functional in mature mouse oocytes/eggs. This is an important gap in our knowledge. However, the authors presentation of this data seems misleading. The authors indicate that miRNAs “do not function during mouse oogenesis” but this seems to go beyond what the published data would warrant. For example, miRNA mediated repression is observed in growing oocytes but not in fully grown oocytes (Ma et al., *Current Biology* 20, 265–270). Similarly, conditional *Dgcr8* mutants use the *zp3* promoter used to drive *Cre* expression may not be expressed in the earliest stages of oogenesis. A more careful and precise description of published data should be presented in the introduction and in the Discussion (page 13) to describe what has been formally shown. Additionally, I’m unsure about whether microRNA analysis (sequencing, Taqman, microarray) was performed in *Dgcr8* or *Dcr* conditional mouse mutants to determine if conditional knockouts had fully abolished mature microRNA in oocytes. This seems important to point out given the Arur lab’s findings herein that microRNAs persist in *Drosha* mutants and in previously published work analyzing *dcr-1* worm mutants in which microRNAs also persist. One possible change could be adding a third model in the abstract (second sentence) that includes the idea that loss of *Dgcr8* or *Dicer* does not fully eliminate microRNA activity. This could also be addressed in the body of the manuscript.

2) The statement that these “observations suggest that a novel miRNA biogenesis pathway may be active in germ cells...” is too far reaching and is not sufficiently supported by the data. It is possible that the pathway is largely the same but with some redundancy at the *Drosha* processing step. The authors should address (in the Discussion section) whether *Dicer* could be acting in the *Drosha*-independent processing of microRNAs since they are closely related proteins (Kwon et al., 2016, *Cell* 164, 81–90) and have both be detected in the nucleus.

3) In the last sentence of the Discussion, the authors cite the conservation in molecular mechanisms from worms to mice as support for their model that microRNAs may play similar roles in mouse and worm oogenesis. However, this rationale is too broad and seems a weak argument for the conclusion of the paper. The final paragraph should be revised to make a more focused and compelling case for how this work contributes to and advances our understanding of microRNA function in oogenesis.

Minor points

- 1) Abstract: the sentence that miRNAs “are also generated in the absence of Drosha function suggesting that these are Drosha independent” is redundant. The authors could state directly that over half of the oogenic microRNAs were found to be generated through an unknown Drosha independent mechanism. Is there an alternative model in which they are not Drosha independent? That is, is it possible that some Drosha activity remains in the drsh-1(ok369) mutants? There is a similar statement in the Introduction, page 4 line 15 that could be similarly revised.
- 2) Introduction: “The Argonaute family of proteins helps stabilize the complex between mature miRNA and the mRNA” The word “stabilize” does not seem to accurately capture the functional significance the role of the Argonaute protein. This should be revised to better describe the role of Ago proteins in the RISC.
- 3) Results, page 5. Most of the first paragraph (lines 9-19) seems far too detailed for the Results section and would be better suited in Methods.
- 4) Results page 6. mir-62 is a ‘mirtron’ that is known to be generated in a Drosha independent mechanism (Ruby et al., Nature. 448: 83–6). This should be addressed.
- 5) Did the authors perform in situ hybridization to detect microRNAs in the somatic gonad or in mature sperm (positive controls)? This should be cited in Results (page 8).
- 6) Results page 9. “cryptic non-coding functions in the DNA” This is confusing and should be explained more clearly.
- 7) DAPI analysis is not included in the Methods section. The authors refer to the membrane ‘marker’, “LAMIN”. Does this refer to antibody labeling? Nothing is cited in the Methods about labeling dissected gonads with antibodies. Also, it’s not clear why it is spelled out and capitalized (LMN-1)? NOP-1 localization is shown in Figure 3 with no reference to this work in Methods.
- 8) Results, page 11. Explain “chromosomal transition or organized oocyte growth” more clearly/thoroughly.
- 9) Results, page 12, first sentence of last paragraph. This sentence is awkwardly phrased and should be revised.
- 10) Discussion, page 14. The sentence “Our work on miRNAs in the C. elegans germline supports the latter model” referring to microRNAs function either as ‘buffering’ regulators or ‘significant control’ type regulators. This seems to be an overstatement of what is shown in this work. The authors do not present data that addresses these two broad models for microRNA function. Further, these are not really competing alternative models but are likely to both be accurate description of different subsets of microRNAs in different cells or in different biological processes or pathways.
- 11) Discussion. The authors should cite the Martinez et al paper (Genome Res. 2008 18: 2005-2015) looking at possible expression of oogenic microRNAs in the somatic gonad.
- 12) Acknowledgements: It does not seem typical to include information about presentations at meetings in acknowledgements.
- 13) Methods. The strain list is incomplete. For example, mir-61/250 (nDf59) is not listed, nor are transgenic strains (strains with Histone 2B mCherry transgenes)

14) Methods. Student's T-test is used throughout this work. In most cases, ANOVA with appropriate post hoc test should be performed.

15) Figure legends and Results—the expression pattern that is observed using in situ hybridization refers to 'cell types' but in most of the hermaphrodite germ 'cells' are in a syncytium. It seems best to refer to regions of the gonad arm and corresponding meiotic stage rather than cell types.

16) The manuscript has numerous examples of errors, imprecise language or inaccurate terms. I've listed many examples below, but likely not all. A full revision for clarity and accuracy is needed.

a. Introduction: change "comprised of" to "composed of"

b. Introduction, page 4 line 2: Ago2 is not 'mutated' but is rather an alternative isoform is present in mouse oocytes.

c. Introduction, page 4 line 7: remove second comma.

d. Introduction, page 4 line 11: change "unveil that" to 'identify'

e. Introduction, last sentence: This is very awkwardly written. The first clause does not seem connected to the second. This should be revised.

f. Results page 5: 'drosha' is spelled out on line 3 but then on line 4, it's explained that drsh-1 will be called drosha. This should be reversed.

g. Results page 5: "absence of pleiotropy" is too vague and doesn't seem to be correct use of the term pleiotropy. The authors should state that they are looking at a stage in the absence of other gross morphological/physiological defects.

h. Results page 7: remove "(Methods)"

i. Results page 7: change "lower copy number", which seems to refer to DNA, to "relatively low levels".

j. The description of the spatial patterns of expression observed using in situ hybridization should be revised through the entire section of the Results. For example, microRNAs are said to "belong to this pattern" or "were named Pattern 2" or "are Pattern 3 miRNAs". This should all be revised for accuracy.

k. Results: "fertilizing oocyte" is used many times. It would be better to describe an 'ovulated oocyte' or a zygote or an early embryo (whatever term is most accurate).

l. Results: microRNA gene names should be lower case (mir-51, not miR-51, when referring to the gene name rather than the small RNA). This needs to be corrected throughout the manuscript and figures. Alleles or genotypes should not be used alone to refer to mutant worms. It is more typical to say mir-35-41 (nDf50) *mutants* (or 'worms' or 'hermaphrodites') when you are referring to the animals rather than to the gene, allele, or microRNA. This section needs to be extensively revised to correct all of the references to mutant worms/germlines. Also, the nDf50 allele removes mir-35-41. However, it is referred to as a "mir-35" deletion in multiple instances. nDf58 removes mir-54-56 but is referred to as mir-54-55 or mir-54 (Figure 3 legend, Methods, p. 25, and Figure 1). Lastly, nDf59 removes both mir-61 and mir-250.

m. Results, page 11. Alvarez-Saavedra et al., Current Biology 20, 367–373 should be cited for mir-35-41 embryonic lethality data.

n. Discussion, page 15. It may be a style decision, but I would recommend revising the start of the first paragraph on page 15 to eliminate the question format.

o. Discussion, page 15. The last sentence in the second paragraph seems unnecessary.

p. Figure legends, Figure 2. Revise "Bright field microscopy reveals in situ expression...and DAPI right highlights germ cell..." for accuracy.

q. Figure legends, Figure 3. Revise "pattern of accumulation" for accuracy. 'Accumulation' seems imprecise.

r. Figure legends, figure 5. Fix "embryo's/animals"

Reviewer #2 (Remarks to the Author):

Arur and colleagues characterize the miRNAome of *C. elegans* oocytes. They discover several miRNAs expressed in the germline during oocyte maturation and identify important roles for oocyte-expressed miRNAs in development. Surprisingly, many of the oocyte-expressed miRNAs are not dependent on the microprocessor complex that normally cleaves the miRNA hairpin from its primary transcript. The results have important implications in our understanding of the miRNA pathway and its roles in germline development. As such, the manuscript will be impactful and should appeal to a broad audience. Some shortcomings, noted below, should be addressed before publication.

Major Points

Mechanistic insight how the drosha-independent miRNAs are processed would add substantially to the impact. I think that most readers would be somewhat satisfied if this was at least explored on some level in the manuscript. Along those lines, here are a few suggestions:

It is possible that some miRNAs are processed via both drosha-dependent and drosha-independent mechanisms (the modest reduction in the levels of some miRNAs in the drosha mutant supports this)? Perhaps the miRNAs identified as drosha-independent are, at least in some instances, only drosha-independent in oocytes. This could be tested by doing the in situ assays in the drosha mutant. It would also be interesting to test those that are expressed in both somatic and germline tissues for dependence on drosha in whole animals in which many of the cells are somatic.

Where is drosha expressed in the germline? Could in situ be done to examine drosha expression in the germline? This might provide some insight into why some miRNAs are not drosha-dependent

The exosome was previously shown to be required for the biogenesis of some miRNAs (Flynt et al. Mol Cell 2010). Is the exosome required for the drosha-independent miRNAs identified here?

Is it possible that some of the drosha-independent miRNAs are mirtrons (miR-62, for example, is an annotated mirtron)? How many of the drosha-independent miRNAs are derived from coding genes? Is there a greater tendency for drosha-independent miRNAs to derive from introns than there is for drosha-dependent miRNAs?

Minor Points and Technical Concerns

Table S1 isn't really necessary since the data is also contained in Table S2. If in S2 the data were sorted by whether the miRNA is expressed in the germline and then indicated as such it would be sufficient.

Page 6, paragraph 2. It should be noted here that miR-62 is a mirtron, which is a nice confirmation of the results.

Page 8, paragraph 2. "Thus, whether a miRNA is dependent on Drosha function or not does not distinguish their localization pattern in the germline." The data seems to suggest that as germ cells progress towards oocytes, there is a tendency for the miRNAs to be drosha-independent.

Figure S13. It is not clear what the shading and Xs within the cells indicate.

How many biological replicates were done in the FirePlex experiment - Tables S1-S2 suggest there were 3 but this does not seem to be described in the methods.

Has the FirePlex probe set been tested across all *C. elegans* miRNAs? Or is there another reason you can be confident that the miRNAs not detected in the germline were simply poorly captured using the approach? Particularly miRNAs that were previously suggested to be in the germline but were not detected here.

Related to the above point - how was RNA isolated? It has been shown that Trizol-like reagents discriminate against certain small RNAs in low cell number preps such as would be the case with dissected germlines (Short Structured RNAs with Low GC Content Are Selectively Lost during Extraction from a Small Number of Cells. *Mol Cell* 2012).

Was a multiple comparison correction applied to the FirePlex data?

We thank the reviewers for the very thoughtful and instructive comments. We have incorporated all of the comments. We believe the reviews have significantly improved the manuscript.

Reviewer # 1 (Remarks to the Author)

In this manuscript, Minogue et al., identify a set of microRNAs that are expressed in the C. elegans hermaphrodite gonad arm, specifically in the germ line. This work is mostly descriptive, focusing on the identification and possible functions of microRNAs in the process of oogenesis and oocyte maturation.

The authors use the FirePlex microRNA assay that allows the researchers to analyze RNA samples for expression of 307 candidate microRNAs found in the miRbase database. Additional work used in situ hybridization and qPCR Taqman assays to validate and further define the spatial expression patterns. Lastly, preliminary functional analysis of oogenic microRNAs was performed using available mutants and a novel approach of using RNAi directed against the pre-microRNA. Results from functional analysis suggest a role for these oogenic microRNAs in the control of meiotic progression in the hermaphrodite germ line.

These results advance our knowledge of the identification and functions of microRNAs in oocyte formation and maturation. This is an important contribution that provides a foundation for future work to delineate specific functions of individual microRNAs. There are a number of points that need to be addressed that would strengthen this manuscript.

Major Comments:

A key point that the authors make is that the role of microRNAs in oogenesis is a central unknown due to conflicting results in different organisms, specifically data in mice that indicates that microRNAs are not functional in mature mouse oocytes/eggs. This is an important gap in our knowledge.

However, the authors presentation of this data seems misleading. The authors indicate that miRNAs “do not function during mouse oogenesis” but this seems to go beyond what the published data would warrant. For example, miRNA mediated repression is observed in growing oocytes but not in fully grown oocytes (Ma et al., Current Biology 20, 265–270).

Similarly, conditional Dgcr8 mutants use the zp3 promoter used to drive Cre expression may not be expressed in the earliest stages of oogenesis. A more careful and precise description of published data should be presented in the introduction and in the Discussion (page 13) to describe what has been formally shown. Additionally, I'm unsure about whether microRNA analysis (sequencing, Taqman, microarray) was performed in Dgcr8 or Dcr conditional mouse mutants to determine if conditional knockouts had fully abolished mature microRNA in oocytes. This seems important to point out given the Arur lab's findings herein that microRNAs persist in Drosha mutants and in previously published work analyzing dcr-1 worm mutants in which microRNAs also persist.

Response: We thank the reviewer for the thoughtful comments. We did not mean to be misleading in our statement that “miRNAs do not function”, this statement reflected the conclusions drawn by the authors in the Ma et al., Current Biology, 265-270 and the accompanying Suh et al., Current Biology, 271-277 manuscripts. Suh et al., Current Biology, 271-277 did not perform a global profiling for miRNAs. However, they analyzed the Dgcr8 and Dicer1 knock out oocytes (at GV stage) for 40 most expressed miRNAs by candidate Taqman based assay (Figure 4 from Suh et al.). They show that 39 of the 40 miRNAs are reduced in both Dicer-/- and Dgcr8 -/- oocytes. They use these data to conclude that Dgcr8 is required for miRNA production in oocytes, but that miRNAs themselves are not required, since despite this depletion, the authors do not see a clear oocyte phenotype or a change in gene expression. We more carefully point out these data in the Introduction.

The statement that these “observations suggest that a novel miRNA biogenesis pathway may be active in germ cells...” is too far reaching and is not sufficiently supported by the data. It is possible that the pathway is

largely the same but with some redundancy at the Drosha processing step. The authors should address (in the Discussion section) whether Dicer could be acting in the Drosha-independent processing of microRNAs since they are closely related proteins (Kwon et al., 2016, Cell 164, 81–90) and have both be detected in the nucleus.

Response: We agree that as written, the statement that “these observations suggest that a novel miRNA biogenesis pathway may be active in germ cells..” does seem strong. We were basing this on three observations that we had made (a) mature miRNAs perdured in *dcr-1(ok247)* homozygous mutants in *C. elegans* (Drake et al., Developmental Cell, 2014: 614-628). Phosphorylated nuclear Dicer in this analysis also did not impact the miRNA production (Drake et al., Developmental Cell, 2014: 614-628). (b) we identified two classes of mature miRNAs that were germline expressed in the current study: drosha dependent class, that is depleted in *drsh-1(ok369)* mutants and an independent class that perdures in the *drsh-1(ok369)* mutants. The latter overlap with miRNAs that perdure in *dcr-1(ok247)*, we present a new Figure S15 to support this. And (c) The Argonoute thought to mediate miRNA-based repression in the germline is ALG-5 (Brown et al., Nucleic Acid Research, 9093-9107). When we overlapped the germline expressed class identified in this study with the ALG-5 dependent class, the *drosha*-independent miRNAs were not present in the ALG-5 IP. Since Drosha, Dicer and ALG-5 are thought to function in the canonical pathway, and the miRNAs were not found to be dependent in any of these three classes, we proposed that there may be a new pathway. As the reviewer suggests, however, there is formal possibility that the miRNAs are redundant for generation at any of these steps. We thus changed the sentence in the Introduction to “These observations suggest that the miRNA biogenesis pathway may not function as a linear pathway as envisioned (with DRSH-1 → DCR-1 → ALG-5) but instead the pathway genes may be redundant with other currently unknow genes, especially at the Drosha processing step”.

In the last sentence of the Discussion, the authors cite the conservation in molecular mechanisms from worms to mice as support for their model that microRNAs may play similar roles in mouse and worm oogenesis. However, this rationale is too broad and seems a weak argument for the conclusion of the paper. The final paragraph should be revised to make a more focused and compelling case for how this work contributes to and advances our understanding of microRNA function in oogenesis.

Response: Thank you for the comment. We take the opportunity suggested by the reviewer and now conclude the Discussion section of the manuscript more focused on how this study contributes to our understanding of miRNA function in oogenesis. We state

“In summary, we identify the functional miRNA repertoire regulating oogenesis during meiosis I in *C. elegans*. Previous work in *C. elegans* showed that loss of Dicer from the germline did not result in any oocyte defects, however, in this case, the miRNAs were perduring thus assaying the direct function of miRNAs in this context was difficult. In mammals, miRNAs are thought to be dispensable for regulating oocyte development^{9,34} and the post-transcriptional gene regulation seems to be funneled through endo-siRNAs to regulate oocyte development¹⁴. Our work suggests that not only does a small repertoire of miRNAs function to regulate oocyte development, but that in addition to canonical pathways for miRNA production, there may be non-canonical and redundant mechanisms that generate functional miRNAs driving oocyte development. We conclude that post-transcriptional gene regulation in oocytes also occurs through specific oocyte-expressed miRNAs, and miRNA function in regulating oocyte development may not be completely dispensable even in the mammalian context”.

Minor Points:

Abstract: the sentence that miRNAs “are also generated in the absence of Drosha function suggesting that these are Drosha independent” is redundant. The authors could state directly that over half of the oogenic microRNAs were found to be generated through an unknown Drosha independent mechanism. Is there an alternative model in which they are not Drosha independent? That is, is it possible that some Drosha activity

remains in the drsh-1(ok369) mutants? There is a similar statement in the Introduction, page 4 line 15 that could be similarly revised.

Response: We have revised the statements throughout the manuscript to minimize redundancy.

It is possible that some *drsh-1(ok369)* activity is perduring due to maternal content. We did test for *drsh-1* mRNA via in situ hybridization during the revision, as suggested by Reviewer # 2, and find that the mRNA itself is very low in the *drsh-1(ok369)* mutant at this stage of development. We present this as new figure: Figure S14. It is, however, possible that there was Drosha activity earlier in development which results in the generation of the miRNAs and a subset of miRNAs are perduring into adulthood. Although, if this was the case, we would still not have observed these in the germline, since the germ cells develop in adulthood, and we assayed dissected germlines for many of our initial assays. But we cannot formally rule out the possibility, thus we acknowledge it.

Introduction: “The Argonaute family of proteins helps stabilize the complex between mature miRNA and the mRNA” The word “stabilize” does not seem to accurately capture the functional significance the role of the Argonaute protein. This should be revised to better describe the role of Ago proteins in the RISC.

Response: We do so in the revision. The sentence reads as follows: “Argonoute family of proteins achieves specific target inhibition as part of the RNA-induced silencing complex (RISC) where they are guided by the miRNA to complementary target sites in mRNAs, and active Ago proteins cleave the targets (Hammond et al., 2000)”.

Results, page 5. Most of the first paragraph (lines 9-19) seems far too detailed for the Results section and would be better suited in Methods.

Response: In hindsight, we completely agree. We move this section to the methods. Thank you.

Results page 6. mir-62 is a ‘mirtron’ that is known to be generated in a Drosha independent mechanism (Ruby et al., Nature. 448: 83–6). This should be addressed

Response: Done. The sentence reads as follows in the revision: “In addition, we also identified a previously described *drosha*-independent miRNA: *mir-62*, in *drosha* mutant germlines. *mir-62* is a mirtron, that resides in intronic regions that form pre-miRNA precursors and are generated through *drosha* independent processing (Ruby et al., 2007). The identification of *mir-62* as *drosha* independent species in this analysis validates our results. Although the mechanisms through which *mir-62* is generated may be distinct than of those miRNAs identified in this study”.

Did the authors perform in situ hybridization to detect microRNAs in the somatic gonad or in mature sperm (positive controls)? This should be cited in Results (page 8).

Response: Somatic gonad components- the distal tip cell (DTC), sheath cells and the spermatheca are displayed in all of the images shown in the figures. During the gonadal dissection and staining process, we do not separate the germline from the somatic gonad or the uterus, thus, while the spermathecal-uterine valve and uterus are not displayed on the images shown, we also imaged them. In our experience, it is almost impossible to not detect the somatic gonadal components, and mature sperm in a gonadal dissection preparation. Thus, each dissection preparation contains all of these cell types. We did not notice any of the somatic gonadal components to be positively stained in the *in situ* hybridization from the miRNA probes. We did visualize somatic gonadal components, as well as sperm in the positive control: U6 in situ hybridization analysis. We cite this more clearly in the text on page 7. Thank you.

Results page 9. “cryptic non-coding functions in the DNA” This is confusing and should be explained more clearly.

Response: Thank you for pointing this out. We clarify it as follows: “..due to an RNA function of the miRNA being tested rather than because the genomic deletion was removing a long-range transcriptional enhancer element, affecting a downstream gene unrelated to the miRNA being tested”.

DAPI analysis is not included in the Methods section. The authors refer to the membrane ‘marker’, “LAMIN”. Does this refer to antibody labeling? Nothing is cited in the Methods about labeling dissected gonads with antibodies. Also, it’s not clear why it is spelled out and capitalized (LMN-1)? NOP-1 localization is shown in Figure 3 with no reference to this work in Methods.

Response: Our apologies. This was a clear oversight on our part. We make these labels clear and clarify the results in the text. We also present Methods used for germline dissection and staining, as well as the antibodies used. Thank you.

Results, page 11. Explain “chromosomal transition or organized oocyte growth” more clearly/thoroughly

Response: We explain this as “where they may function to regulate the transition of chromosomes from pachytene or diplotene stage of meiosis or they may function to organize the linear row of growing diplotene oocytes (Figure 4H)”.

Results, page 12, first sentence of last paragraph. This sentence is awkwardly phrased and should be revised.

Response: We rephrased it as following: “Together these data reveal that oocyte-expressed miRNAs localize to distinct regions of the gonad defined by the chromosomal state of the germ cells (for example, pachytene stage, or diakinetik stage). Functional analysis demonstrates that the oocyte-expressed miRNAs regulate specific oocyte progression and developmental phenotypes in their region of expression. The oogenic defects result in reduced fertility of the animals, as assessed by brood size analysis”.

Discussion, page 14. The sentence “Our work on miRNAs in the C. elegans germline supports the latter model” referring to microRNAs function either as ‘buffering’ regulators or ‘significant control’ type regulators. This seems to be an overstatement of what is shown in this work. The authors do not present data that addresses these two broad models for microRNA function. Further, these are not really competing alternative models but are likely to both be accurate description of different subsets of microRNAs in different cells or in different biological processes or pathways.

Response: We were attempting to highlight that a small subset of miRNAs is functional during oogenesis, rather than a large number of miRNAs, which seems to be the normal expectation. However, we completely agree with the reviewer that (a) the two models may not be distinct at all and (b) our results at this time do not provide concrete evidence and the discussion was more speculative. In light of this, we have removed this entire section, and instead focus more on the relevance of the work to the field.

Discussion. The authors should cite the Martinez et al paper (Genome Res. 2008 18: 2005-2015) looking at possible expression of oogenic microRNAs in the somatic gonad.

Response: Thank you for the suggestion. We do so for miR-246 in the revision. We did not detect this miRNA in the dissected gonad in the genomic analysis. It is likely that miR-246 is more abundant in the spermathecal-uterine valve and the uterine tissue rather than the distal somatic gonad.

Acknowledgements: It does not seem typical to include information about presentations at meetings in acknowledgements.

Response: We are happy that the reviewer noticed. It is not typical. We acknowledged these presentations because we obtained some great feedback from the field as the study was in progress. We specifically spell out what we are thankful for.

Methods. The strain list is incomplete. For example, mir-61/250 (nDf59) is not listed, nor are transgenic strains (strains with Histone 2B mCherry transgenes).

Response: We now do so.

Methods. Student's T-test is used throughout this work. In most cases, ANOVA with appropriate post hoc test should be performed.

Response: We re-did the statistical analysis with one-way ANOVA with the Bonferroni's correction. We have updated the legends to reflect this.

Figure legends and Results—the expression pattern that is observed using in situ hybridization refers to 'cell types' but in most of the hermaphrodite germ 'cells' are in a syncytium. It seems best to refer to regions of the gonad arm and corresponding meiotic stage rather than cell types.

Response: Done.

The manuscript has numerous examples of errors, imprecise language or inaccurate terms. I've listed many examples below, but likely not all. A full revision for clarity and accuracy is needed.

Response: We have carefully read through and revised the manuscript to eliminate the usage of any such terms. Thank you.

a. Introduction: change "comprised of" to "composed of"

Response: Done.

b. Introduction, page 4 line 2: Ago2 is not 'mutated' but is rather an alternative isoform is present in mouse oocytes.

Response: Done.

c. Introduction, page 4 line 7: remove second comma.

Response: Done.

d. Introduction, page 4 line 11: change "unveil that" to 'identify'

Response: Done.

e. Introduction, last sentence: This is very awkwardly written. The first clause does not seem connected to the second. This should be revised.

Response: Done.

f. Results page 5: 'drosha' is spelled out on line 3 but then on line 4, it's explained that drsh-1 will be called drosha. This should be reversed.

Response: Done.

g. Results page 5: “absence of pleiotropy” is too vague and doesn’t seem to be correct use of the term pleiotropy. The authors should state that they are looking at a stage in the absence of other gross morphological/physiological defects.

Response: Done.

h. Results page 7: remove “(Methods)”

Response: Done.

i. Results page 7: change “lower copy number”, which seems to refer to DNA, to “relatively low levels”.

Response: Done.

j. The description of the spatial patterns of expression observed using in situ hybridization should be revised through the entire section of the Results. For example, microRNAs are said to “belong to this pattern” or “were named Pattern 2” or “are Pattern 3 miRNAs”. This should all be revised for accuracy.

Response: We have updated these. Thank you.

k. Results: “fertilizing oocyte” is used many times. It would be better to describe an ‘ovulated oocyte’ or a zygote or an early embryo (whatever term is most accurate).

Response: Thank you for the comment. We debated throughout the duration of the study, as to the exact nomenclature. This is a stage of the oocyte after ovulation, once it is ovulated in the spermatheca, but before it is an embryo (since there is no egg shell present, which is likely why we obtained the positive staining result). Thus, we did not feel comfortable calling this stage “early embryo” as there was no egg shell yet. We did not feel comfortable calling it ovulated oocyte, since the oocyte is likely undergoing meiosis II/ entering mitosis. Thus, we named it fertilizing oocyte. For accuracy however, we have changed this to “oocyte undergoing fertilization”.

*Results: microRNA gene names should be lower case (mir-51, not miR-51, when referring to the gene name rather than the small RNA). This needs to be corrected throughout the manuscript and figures. Alleles or genotypes should not be used alone to refer to mutant worms. It is more typical to say mir-35-41(nDf50) *mutants* (or ‘worms’ or ‘hermaphrodites’) when you are referring to the animals rather than to the gene, allele, or microRNA. This section needs to be extensively revised to correct all of the references to mutant worms/germlines. Also, the nDf50 allele removes mir-35-41. However, it is referred to as a “mir-35” deletion in multiple instances. nDf58 removes mir-54-56 but is referred to as mir-54-55 or mir-54 (Figure 3 legend, Methods, p. 25, and Figure 1). Lastly, nDf59 removes both mir-61 and mir-250.*

Response: Thank you. We do so throughout and keep the nomenclature consistent.

Results, page 11. Alvarez-Saavedra et al., Current Biology 20, 367–373 should be cited for mir-35-41 embryonic lethality data.

Response: Done.

Discussion, page 15. It may be a style decision, but I would recommend revising the start of the first paragraph on page 15 to eliminate the question format.

Response: We did so, and revised the first paragraph of that section. Thank you.

Discussion, page 15. The last sentence in the second paragraph seems unnecessary.

Response: We revised the ending of the second paragraph on Page 15, and the sentence was eliminated.

Figure legends, Figure 2. Revise “Bright field microscopy reveals in situ expression...and DAPI right highlights germ cell...” for accuracy.

Response: Done.

Figure legends, Figure 3. Revise “pattern of accumulation” for accuracy. ‘Accumulation’ seems imprecise.

Response: Done.

Figure legends, figure 5. Fix “embryo’s/animals”

Response: Done.

Reviewer # 2 (Remarks to the Author)

Arur and colleagues characterize the miRNAome of C. elegans oocytes. They discover several miRNAs expressed in the germline during oocyte maturation and identify important roles for oocyte-expressed miRNAs in development. Surprisingly, many of the oocyte-expressed miRNAs are not dependent on the microprocessor complex that normally cleaves the miRNA hairpin from its primary transcript. The results have important implications in our understanding of the miRNA pathway and its roles in germline development. As such, the manuscript will be impactful and should appeal to a broad audience. Some shortcoming, noted below, should be addressed before publication.

Major Points

Mechanistic insight how the drosha-independent miRNAs are processed would add substantially to the impact. I think that most readers would be somewhat satisfied if this was at least explored on some level in the manuscript. Along those lines, here are a few suggestions:

It is possible that some miRNAs are processed via both drosha-dependent and drosha-independent mechanisms (the modest reduction in the levels of some miRNAs in the drosha mutant supports this)? Perhaps the miRNAs identified as drosha-independent are, at least in some instances, only drosha-independent in oocytes. This could be tested by doing the in situ assays in the drosha mutant. It would also be interesting to test those that are expressed in both somatic and germline tissues for dependence on drosha in whole animals in which many of the cells are somatic.

Response: Thank you for the comment. We agree that mechanistic insight into how *drosha*-independent miRNAs are processed would be impactful, especially if the miRNAs were also *drosha*-independent outside of the germline. To that end, we had in fact conducted all the Taqman assays in the first iteration of the manuscript on whole animals from wild type as well as *drosha(ok369)* mutants, staged at 18 hours past L4 stage of development. We find, using the Taqman assays on whole animals, that the *drosha*-independent miRNAs are generated in the absence of Drosha activity (Figure 1). We restructure the sentence in the results and greatly clarify this point.

Where is drosha expressed in the germline? Could in situ be done to examine drosha expression in the germline? This might provide some insight into why some miRNAs are not drosha-dependent

Response: Thank you for the comment. We present a Drosha *in situ* hybridization assay in wild type and mutant *drosha(ok369)* germline as a new Figure S14. We find that Drosha is expressed at the mRNA level throughout the germline in wild type. Given that both the *drosha*-dependent and independent miRNAs have overlapping patterns of expression in the germline, per the *in situ* hybridization analysis using the LNA probes, we don't think that lack of Drosha expression explains the presence of the *drosha*-independent species. The identification of *drosha*-independent class is highly intriguing, and we very much hope that the field finds our analysis useful in delineating these pathways in the future.

The exosome was previously shown to be required for the biogenesis of some miRNAs (Flynt et al. Mol Cell 2010). Is the exosome required for the drosha-independent miRNAs identified here?

Response: We did not specifically assay the role of *dis-3* or any of the exosomal components in the generation of the *drosha*-independent miRNAs. It is however likely that members of the exosomal complex may be involved in processing *drosha*-independent species. We did analyze the genomic locations of each of the oocyte miRNAs to assess whether all the *drosha*-independent species were intronic. We present this as new Figure S16. We find that, there was no clear difference in the *drosha*-dependent and independent species in terms of their genomic location which would allow us to make a prediction as to their biogenesis. For example, the exosome complex processes mirtronic miRNAs. Based on this we could have tested intronic miRNAs, that were not clear mirtrons, but could be hypothesized to be funneled through the splicing / exosome pathway. However, we found that the *drosha*-dependent class comprised of miR-35-41 is also intronic, although not mirtronic, like the other *drosha*-independent class. Thus, we believe that there may be multiple mechanisms that may mediate the generation of the *drosha*-independent class, although given the speculative nature of this argument, we do not discuss it.

Is it possible that some of the drosha-independent miRNAs are mirtrons (miR-62, for example, is an annotated mirtron)? How many of the drosha-independent miRNAs are derived from coding genes? Is there a greater tendency for drosha-independent miRNAs to derive from introns than there is for drosha-dependent miRNAs?

Response: That's a great question. We did not present the data on the genomic localization of the miRNAs in the first round of the manuscript, since we did not find a clear pattern. We present it in the revised manuscript as new Figure S16. miR-62 is the only clear mirtron, based on the canonical definition of the mirtron. However, we find that several of the *drosha*-independent and dependent miRNAs are either intergenic or intronic in nature. And in one case, intron 11 of *gcn-1* gene carries miR-72-5p, which is *drosha*-dependent and the same intron also carries miR-64-66 cluster, which are *drosha*-independent while mir-229 which is part of the mir-64-66 cluster is *drosha*-dependent. Thus, we did not find a clear pattern to help differentiate the two classes based on their genomic localization.

Minor Points and Technical Concerns

Table S1 isn't really necessary since the data is also contained in Table S2. If in S2 the data were sorted by whether the miRNA is expressed in the germline and then indicated as such it would be sufficient

Response: We agree. We now just indicate the data from Table S2 in Table S1.

Page 6, paragraph 2. It should be noted here that miR-62 is a mirtron, which is a nice confirmation of the results.

Response: We agree. Thank you for the suggestion. We now do so.

Page 8, paragraph 2. “Thus, whether a miRNA is dependent on Drosha function or not does not distinguish their localization pattern in the germline.” The data seems to suggest that as germ cells progress towards oocytes, there is a tendency for the miRNAs to be drosha-independent.

Response: We agree, that there may be a trend for *drosha*-independent miRNAs to express in arrested and maturing oocytes more than in pachytene germ cells, for example. The statement, however, was made specifically in the context of Pattern 2 miRNAs, which displayed expression in diakinetik and maturing oocytes, where three were *drosha*-dependent and four were *drosha*-independent. In the revised text however, we add the trend in the final section of the *in situ* hybridization section. Thank you.

Figure S13. It is not clear what the shading and Xs within the cells indicate.

Response: We clarify the legend. Thank you.

How many biological replicates were done in the FirePlex experiment - Tables S1-S2 suggest there were 3 but this does not seem to be described in the methods.

Response: Three biological replicates were done for this experiment. We clarify this in the methods. Thank you.

Has the FirePlex probe set been tested across all C. elegans miRNAs? Or is there another reason you can be confident that the miRNAs not detected in the germline were simply poorly captured using the approach? Particularly miRNAs that were previously suggested to be in the germline but were not detected here.

Response: Fireplex analysis has been performed on *C. elegans* miRNAs. miRNAs that were germline ‘negative’ in our assay have been positively identified in worms using Fireplex in the following papers: Ren and Ambros, PNAS, 2015, 18:2366-2375 (assayed let-7 family comprised of: miR-48, miR-84, miR-241 and let-7). Zinovyeva, Bousaker, Simard, Hammell and Ambros, Plos Genetics, 2014 (here miRNAs from L1 and L2 larvae were assayed by Fireplex analysis from *alg-1* and *alg-2* mutants). And work from Craig Hunter’s group: Shiu, Zhuang and Hunter, Methods in Molecular Biology, 2014, 1173: 71-87 shows that the Fireplex technology is magnitudes of order more sensitive at assaying small RNAs (miRNA, siRNAs and piRNAs) than next generation sequencing. So, we do believe that the miRNAs that come up as “background” in this study are likely either very low in expression and thus difficult to discern from background or absent. The one miRNA previously detected to be germline expressed and validated by *in situ* hybridization: miR-250, in McEwen et al., was not detected in this method. We performed the *in situ* hybridization to confirm the previous data, and did not detect anything above background in our analysis (we present this in the revised manuscript in discussion as data not shown). At this time, we cannot explain this discrepancy. But we do think that our analysis identified most functional oogenic miRNAs.

Related to the above point - how was RNA isolated? It has been shown that Trizol-like reagents discriminate against certain small RNAs in low cell number preps such as would be the case with dissected germlines (Short Structured RNAs with Low GC Content Are Selectively Lost during Extraction from a Small Number of Cells. Mol Cell 2012).

Response: Thank you for the comment. We agree, that using Trizol may generate issues. Thus, for Fireplex analysis, the total RNA for the Fireplex analysis was extracted using the Nucleospin XS reagent from Macherey-Nagel. Specifically, the total RNA was extracted in the miRNA homogenate additive consisting of 2M Sodium Acetate, pH. 4, and extracted by phenol chloroform. Small RNAs were extracted by size fractionation from each genotype. We clarify this in the methods section. We find that this extraction method is also very sensitive to extraction from small sample material such as dissected germlines. For subsequent Taqman validations, we have used both Nucleospin miRNA homogenate additive or Trizol interchangeably (while keeping them consistent across one experiment) with little change in the data.

Was a multiple comparison correction applied to the FirePlex data?

Response: Yes. We used the adjusted p-values and the adjusted p-values underwent multiple-comparison correction using Bonferroni. We also add this to the methods for clarification.

REVIEWERS' COMMENTS:

Reviewer #1 (Remarks to the Author):

The authors addressed all concerns and comments from the reviewers. The manuscript is much improved by the revisions.

There are some minor errors in the revised manuscript

-page 5, line 75 fix "unknow" genes

-page 6, line 111, fix "two data set"

-page 7, paragraph starting at 118. "germline" is used in many instances where "gonad" or gonad arm is more appropriate. The worm gonad arms were dissected, not the germlines.

-Results. References to Materials and Methods in parentheses seems unnecessary. (line 122,205,208.

-line 258/259, it is inconsistent to use quotation marks around Pattern: "Pattern 3" and "Pattern 4"

-line 585 correct "mir-6;mir-2501 (nDf59) to mir-61 mir-250.

-line 631 change "mir-61;mir-250(nDf59" to "mir-61 mir-250(nDf59)"

Reviewer #2 (Remarks to the Author):

The authors adequately addressed the shortcoming noted in my original comments and I believe the manuscript is now ready for publication. Just one minor point: the complex use of shading in the new figure S16 is not described in a key or in the figure legend.

Reviewer # 1 (Remarks to the Author)

The authors addressed all concerns and comments from the reviewers. The manuscript is much improved by the revisions.

There are some minor errors in the revised manuscript

- page 5, line 75 fix "unknow" genes
- page 6, line 111, fix "two data set"
- page 7, paragraph starting at 118. "germline" is used in many instances where "gonad" or gonad arm is more appropriate. The worm gonad arms were dissected, not the germlines.
- Results. References to Materials and Methods in parentheses seems unnecessary. (line 122,205,208.
- line 258/259, it is inconsistent to use quotation marks around Pattern: "Pattern 3" and "Pattern 4"
- line 585 correct "mir-6;mir-2501 (nDf59) to mir-61 mir-250.
- line 631 change "mir-61;mir-250(nDf59" to "mir-61 mir-250(nDf59)"

Response: We thank the reviewer for all of the previous comments which greatly improved the manuscript. We have incorporated all of the suggested edits and more carefully read through and improved the accuracy of nomenclature in the manuscript.

Reviewer # 2 (Remarks to the Author)

The authors adequately addressed the shortcoming noted in my original comments and I believe the manuscript is now ready for publication. Just one minor point: the complex use of shading in the new figure S16 is not described in a key or in the figure legend.

Response: Thank you very much. Thank you also for the previous round of reviews, the comments greatly improved the manuscript. We clearly spell out the shaded colors in the Figure legend for Supplementary Figure 16.

We thank both the reviewers for the thoughtful comments.